# Statistical predictability of the Arctic sea ice volume anomaly: identifying predictors and optimal sampling locations

Leandro Ponsoni[1,2], François Massonnet[1,2], David Docquier[3], Guillian Van Achter[1], and Thierry Fichefet[1]

[1]Georges Lemaître Centre for Earth and Climate Research (TECLIM), Earth and Life Institute, Université catholique de Louvain, Louvain-la-Neuve, Belgium
[2]Fonds de la Recherche Scientifique – FNRS, Belgium
[3]Rossby Centre, Swedish Metereological and Hydrological Institute, Norrköping, Sweden

*Correspondence to:* Leandro Ponsoni (leandro.ponsoni@uclouvain.be)

**Abstract.** This work evaluates the statistical predictability of the Arctic sea ice volume (SIV) anomaly – here defined as the detrended and deseasonalized SIV – on the interannual time scale. To do so, we made use of six datasets, from three different atmosphere-ocean general circulation models, with two different horizontal grid resolutions each. Based on these datasets, we have developed a statistical empirical model which in turn was used to test the performance of different predictor variables, as well as to identify optimal locations from where the SIV anomaly could be better reconstructed and/or predicted. We tested the hypothesis that an ideal sampling strategy characterized by only a few optimal sampling locations can provide *in situ* data for statistically reproducing and/or predicting the SIV interannual variability. The results showed that, apart from the SIV itself, the sea ice thickness is the best predictor variable, although total sea ice area, sea ice concentration, sea surface temperature, and sea ice drift can also contribute to improving the prediction skill. The prediction skill can be enhanced further by combining several predictors into the statistical model. Applying the statistical model with predictor data from four well-placed locations is sufficient for reconstructing about 70% of the SIV anomaly variance. As suggested by the results, the four first best locations are placed at the transition Chukchi Sea–Central Arctic–Beaufort Sea (158.0°W, 79.5°N), near the North Pole (40.0°E, 88.5°N), at the transition Central Arctic–Laptev Sea (107.0°E, 81.5°N), and offshore the Canadian Archipelago (109.0°W, 82.5°N), in this respective order. Adding further to six well placed locations, which explains about 80% of the SIV anomaly variance, the statistical predictability does not substantially improve taking into account that ten locations explain about 84% of that variance. An improved model horizontal resolution allows a better trained statistical model so that the reconstructed values approach better to the original SIV anomaly. On the other hand, if we inspect the interannual variability, the predictors provided by numerical models with lower horizontal resolution perform better when reconstructing the original SIV variability. We believe that this study provides recommendations for the ongoing and upcoming observational initiatives, in terms of an Arctic optimal observing design, for studying and predicting not only the SIV values but also its interannual variability.

# 1 Introduction

The ongoing melting of the Arctic sea ice observed in the last decades (e.g., Chapman and Walsh, 1993; Parkinson et al., 1999; Rothrock et al., 1999; Parkinson and Cavalieri, 2002; Zhang and Walsh, 2006; Stroeve et al., 2007, 2012; Notz and Stroeve, 2016; Petty et al., 2018), associated with the respective reductions in total sea ice area (SIA) and volume (SIV), had significant
impact on climate processes at global and regional scales. Globally, the sea ice depletion is reported to impact aspects of the weather at low- and mid-latitude regions, by means of both oceanographic (Drijfhout, 2015; Sévellec et al., 2017) and atmospheric teleconnections (Serreze et al., 2007; Overland and Wang, 2010), including the increased occurrence of extreme events (Francis and Vavrus, 2012; Tang et al., 2013; Screen and Simmonds, 2013; Cohen et al., 2014). Regionally, high-trophic predators such as seabirds (Grémillet et al., 2015; Amélineau et al., 2016) and mammals (Laidre et al., 2008; Lydersen et al.,
2017; Wilder et al., 2017; Pagano et al., 2018; Brown et al., 2016) are adapting their foraging behavior and dietary preferences. At the same time, native communities experienced a disturbance in subsistence activities like fishing, crabbing and hunting (Nuttall et al., 2005). Other pressing local issues are also bringing important implications for the Arctic countries such as the opening of new ship routes (Lindstad et al., 2016), the development of the tourism industry (Handorf, 2011) and the mineral resource extraction (Gleick, 1989).

Since this intense sea ice loss is projected to continue throughout the twenty-first century (e.g., Burgard and Notz, 2017), the interest of the scientific community and policy makers on the sea ice variability and predictability is increasing exponentially, mainly in terms of SIV. The SIV is a primary sea ice diagnostic because it accounts for the total mass of sea ice. However, *in situ*- and/or satellite-based estimates of SIV are still sparse and temporally sporadic (Lindsay, 2010; Tilling et al., 2018). Due to this lack of continuous long-term observations, there is no clear answer to the question of whether or not this decline
in sea ice is affecting the interannual variability of the pan-Arctic SIV, and the other way around. Nevertheless, recent model analyses suggests the latter (Van Achter et al., 2019). Despite the fact that current atmosphere-ocean general circulation models (AOGCMs), including their respective sea ice component, are increasingly complex, being sometimes used to estimate the quality of global observational datasets (Massonnet et al., 2016), *in situ* observations are still required for a more comprehensive model validation and also for assimilation purposes.

In order to respond to the need of having an improved observational system for better understanding the SIV variability, but at the same time minimize the costs required to do so, this work raises the hypothesis that "an ideal sampling strategy characterized by only a few optimal sampling locations can provide *in situ* data for statistically reproducing and/or predicting the SIV interannual variability". To test this hypothesis, this study follows three main directions. First, we propose a statistical empirical model for predicting the SIV. Since we are mainly interested in predicting the interannual variability rather than the
seasonal cycle and the long-term trends, we will focus on the SIV without these two components – hereafter defined as SIV anomaly. Second, we investigate the performance of a set of ocean- and ice-related predictor variables as input into the empirical model. Third, we intend to localize a reduced number of optimal sampling locations from where the predictor variables could be systematically sampled using oceanographic moorings and/or buoys. Sampling *in situ* data at optimal locations or, in order words, by collecting data at locations in which most of the pan-Arctic SIV anomaly variability is captured by the predictor

variables, makes it much more feasible to sustain a long-term programme of operational oceanography both from logistical and financial points of view.

To the best of the authors' knowledge, this study is the first to apply an empirical statistical model for supporting an optimal observing system of the pan-Arctic SIV anomaly, albeit a similar study was conducted by Lindsay and Zhang (2006) a decade ago. However, they focused on the predictability of averaged Arctic sea ice thickness, based their results on a single model approach, as well as considered two predetermined sampling locations. Other previous works also applied statistical empirical models for predicting a range of Arctic sea ice properties (e.g., sea ice extent, area and concentration), for lead periods of up to one year, at regional and/or pan-Arctic scales (Walsh, 1980; Barnett, 1980; Johnson et al., 1985; Drobot and Maslanik, 2002; Drobot et al., 2006; Lindsay et al., 2008; Chevallier and Salas-Mélia, 2012; Grunseich and Wang, 2016; Yuan et al., 2016). Unlike the statistical prediction of sea ice extent and area, which have longer and more reliable observational records allowing statistical models to be built on these data, the statistical prediction of SIV necessarily requires information from models. *In situ* measurements of sea ice thickness needed for calculating the SIV are far too expensive, while satellite campaigns such as ICESat (Kwok et al., 2007), CryoSat-2 (Kwok and Cunningham, 2015), and SMOS (Tian-Kunze et al., 2014; Kaleschke et al., 2016) present well-known limitations for sampling the sea ice thickness during the warmer months.

Thus, even though we claim that *in situ* observations are crucial for understanding the SIV variability, our study makes use of outputs from three AOGCMs. This is the only way to have continuous and well distributed data of the predictand and some predictor variables, such as sea ice thickness. The AOGCMs used in this work are cutting edge in terms of model physics and resolution (Haarsma et al., 2016) so that they fairly represent the thermodynamic and dynamic processes linking predictors to predictand. The use of three different models attempts to assess the model dependence of our results.

To fully address the three overall directions and the hypothesis described above, this study is guided by the following open questions: (i) What is the performance of different pan-Arctic predictors for predicting pan-Arctic SIV anomalies? (ii) What are the best locations for *in situ* sampling of predictor variables to optimize the statistical predictability of SIV anomalies in terms of reproducibility and variability? (iii) How many optimal sites are needed for explaining a substantial amount (e.g., 70% – an arbitrarily chosen threshold) of the original SIV anomaly variance? (iv) Are the results model dependent, in particular, are they sensitive to horizontal resolution?

## 2 Data and methods

### 2.1 Model outputs

This work follows a multi-model approach. It takes advantage of six coupled historical runs from three different AOGCMs (each with two horizontal grid resolutions), all conducted within the context of the High Resolution Model Intercomparison Project (HighResMIP; Haarsma et al. (2016)). The HighResMIP is endorsed by the Coupled Model Intercomparison Project 6 (CMIP6; Eyring et al. (2016)) and its main goal is to systematically study the role of horizontal resolution in the simulation of the climate system. These historical coupled experiments, referred to as "hist-1950" (Haarsma et al., 2016), start in the early 1950s and span for about 65 years until mid-2010s. They are not pegged to observed conditions and their initial state is achieved

by control coupled experiments referred to as "control-1950", also produced in the context of HighResMIP. "control-1950" runs are CMIP6-like pre-industrial control ("piControl") experiments, but using fixed 1950s forcing (Haarsma et al., 2016) rather than 1850s forcing as in "piControl" (Eyring et al., 2016). The forcing consists of greenhouse gases (GHG), including $O_3$ and aerosol loading provided by a 10-year mean climatology for the 1950s (Haarsma et al., 2016). A full description of the

GHG concentrations used by CMIP6 and HighResMIP is presented in Meinshausen et al. (2017).

The AOGCMs are: the version 1.1 of the Alfred Wegener Institute Climate Model (AWI-CM; 201; Rackow et al. (2018)), the European Centre for Medium-Range Weather Forecasts Integrated Forecast System (ECMWF-IFS) cycle 43r1 (Roberts et al., 2018), and the Global Coupled 3.1 configuration of the Hadley Centre Global Environmental Model 3 (HadGEM3-GC3.1; Roberts et al. (2019)).

A comprehensive comparison including these three models and their respective specifications are presented by Docquier et al. (2019). In short, AWI-CM is composed by the European Centre/Hamburg version 6.3 (ECHAM6.3) atmospheric model and by the version 1.4 of the Finite Element Sea ice-Ocean Model (FESOM; Wang et al. (2014); Sein et al. (2016)). ECMWF-IFS is a hydrostatic, semi-Lagrangian, semi-implicit dynamical-core atmospheric model, while the ocean and ice components are composed by the version 3.4 of the Nucleus for European Modelling of the Ocean model (NEMO; Madec (2008)) and

version two of the Louvain-la-Neuve Sea-Ice Model (LIM2; Fichefet and Maqueda (1997)), respectively. Finally, HadGEM3-GC3.1 is built up with the same ocean model than ECMWF-IFS (NEMO; Madec et al. (2017)), but version 3.6, the atmospheric Unified Model (UM; Cullen (1993)) and the version 5.1 of the Los Alamos sea-ice model (CICE; Hunke et al. (2015)). Hereafter the models are simply referred to as AWI, ECMWF, and HadGEM3.

The two configurations from the same model keep the parameters identical, except for the resolution-dependent parameteri-

zations (Docquier et al., 2019). In terms of ocean–sea ice grid, both AWI versions (data source: Semmler et al. (2017a, b)) use a mesh grid with varying resolution, in which dynamically active regions have higher resolution. The low-resolution version (AWI-LR) has a nominal resolution of 250 km (e.g., 129 km at 50.0°N and 70 km at 70.0°N), while the high-resolution version (AWI-HR) has nominal resolution of 100 km (e.g., 67 km at 50.0°N and 36 km at 70.0°N). Nevertheless, both versions have a similar resolution of ∼25 km in the Arctic Ocean. For a better understanding of AWI's grid, the reader is referred to Sein et al.

(2016) (their Fig. 4a,b). Both ECMWF (data source: Roberts et al. (2017a, b)) and HadGEM3 (data source: Roberts (2017a, b)) adopt the tripolar ORCA grid (Madec and Imbard, 1996). The configurations with coarser resolution (ECMWF-LR and HadGEM3-LL) use ORCA1 with a resolution of 1°, while the versions with higher horizontal grid spacing (ECMWF-HR and HadGEM3-MM) use ORCA025 with a resolution of 0.25°. In terms of time resolution, our results are all based on monthly outputs from these model simulations.

For the three models, the SIV time series from the versions with a coarser horizontal grid present higher mean values compared to their respective finer resolution versions (Fig. 1a). The differences between the two versions are about $4.52 \times 10^3$ km$^3$ and $2.56 \times 10^3$ km$^3$ for AWI and HadGEM3, but much larger to ECMWF ($26.17 \times 10^3$ km$^3$). The standard deviations (STD) from the SIV anomalies indicate that interannual variabilities are also higher for the coarser grid versions (Fig. 1b). The difference between coarser and higher resolutions for AWI, ECMWF, and HadGEM3 are $0.30 \times 10^3$ km$^3$, $1.78 \times 10^3$ km$^3$, and

$0.43 \times 10^3$ km$^3$. We recall that the term anomaly in this work refers to the detrended and deseasonalized time series. In practical

terms, the anomaly is calculated by excluding the individual trend provided by a second-order polynomial fit of each individual month.

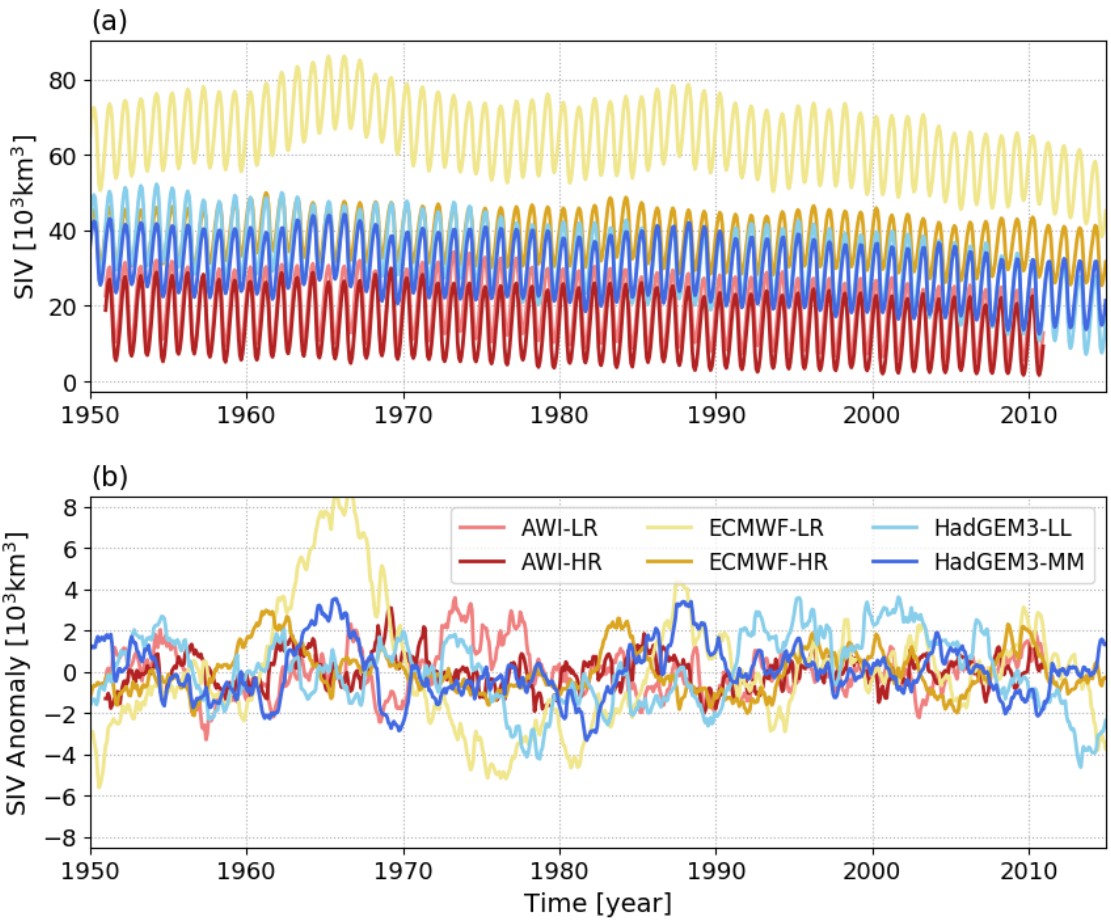

**Figure 1.** Sea ice volume time series from the six model configurations used in this work: (a) absolute values and (b) anomalies in which the long-term trends and the seasonal cycles were subtracted from the original time series.

## 2.2 Potential predictors

In this section, we identify potential predictor variables for using as input into the empirical statistical model that predicts SIV anomalies. Apart from the condition that all predictor variables could be regularly sampled from observational platforms in the real-world, we only pre-selected variables which have the potential to impact the sea ice through dynamic and/or thermo-dynamic processes. Overall, two categories of predictors are tested: integrated variables, intrinsically represented by a single pan-Arctic time series, and predictors represented by several gridded time series of the same variable. Here, predictor variables are also considered in terms of their anomaly.

In total, a set of seven predictors are considered for this preliminary inspection. Three of them are integrated variables, that are: pan-Arctic SIV itself, pan-Arctic sea ice area (SIA) and Atlantic basin ocean heat transport (OHT) estimated at 60.0°N. The other four predictors are variables organized in a gridded format, that are: sea ice thickness (SIT), sea ice concentration (SIC), sea surface temperature (SST) and sea ice drift (Drift). Fig. 2 shows an example case (AWI-LR) in which the predictand (SIV) is compared against the two intrinsic pan-Arctic predictors (Fig. 2a,b) and against the four gridded predictors (Fig. 2c–j). As a first test, we inspect the performance of pan-Arctic predictors by estimating their lag-0 correlation against the predictand. The correlation coefficients showed in the second (SIA) and third (OHT) columns of Table 1 indicate that SIA is a valid predictor for all model outputs, while OHT is significantly correlated only for the low-resolution versions of the models.

To obtain the same first assessment to the other predictors, the gridded values are reduced to their pan-Arctic average. To do so, the time series are twice normalized: first, by the grid area of each grid cell and, second, by the correlation maps with the predictand (Drobot et al., 2006), as shown in Fig. 2e,f,i,j. In the second normalization, the significant correlation coefficients from the different grid cells are used as normalizing factors (as it is the grid-cell area in the first normalization). The idea behind this second normalization is to take advantage of the correlations between predictand and predictors since the former is not necessarily correlated to the latter over the entire Arctic domain. Note in the maps that insignificant correlation coefficients are set to zero (white regions) so that they do not weigh in the normalization (Fig. 2e,f,i,j). The red lines in Fig. 2c,d,g,h show the respective SIT, SIC, SST and Drift anomalies reduced to their pan-Arctic averages, which are in turn significantly correlated with the predictand in all model outputs (Table 1).

**Table 1.** Lag-0 correlation coefficient estimated between the predictand (SIV anomaly) and a set of pan-Arctic potential predictors: SIA, OHT, SIT, SIC, SST, and Drift. The correlation coefficients between OHT and SIV anomaly for the high-resolution model versions are not shown since only statistically significant coefficients are displayed in the table. Regional predictors (SIT, SIC, SST and Drift) are represented by pan-Arctic averages. As for the predictand, all predictors are used with monthly time-resolution and in terms of their anomaly.

| Models | Predictors | | | | | |
|---|---|---|---|---|---|---|
| | SIA | OHT | SIT | SIC | SST | Drift |
| AWI-LR | 0.64 | -0.08 | 0.86 | 0.29 | -0.57 | 0.15 |
| AWI-HR | 0.69 | – | 0.89 | 0.26 | -0.50 | 0.31 |
| ECMWF-LR | 0.20 | 0.28 | 0.95 | 0.31 | -0.12 | -0.20 |
| ECMWF-HR | 0.24 | – | 0.63 | 0.37 | -0.22 | -0.13 |
| HadGEM3-LL | 0.63 | -0.33 | 0.91 | 0.71 | -0.54 | -0.28 |
| HadGEM3-MM | 0.63 | – | 0.94 | 0.62 | -0.45 | -0.31 |

## 2.3 Statistical empirical models

The basis of our statistical empirical model (SEM) is a multiple linear regression model where the time series of the dependent variable ($y$) could be described as a function of the time series of the independent explanatory variables ($x_i$), as follows:

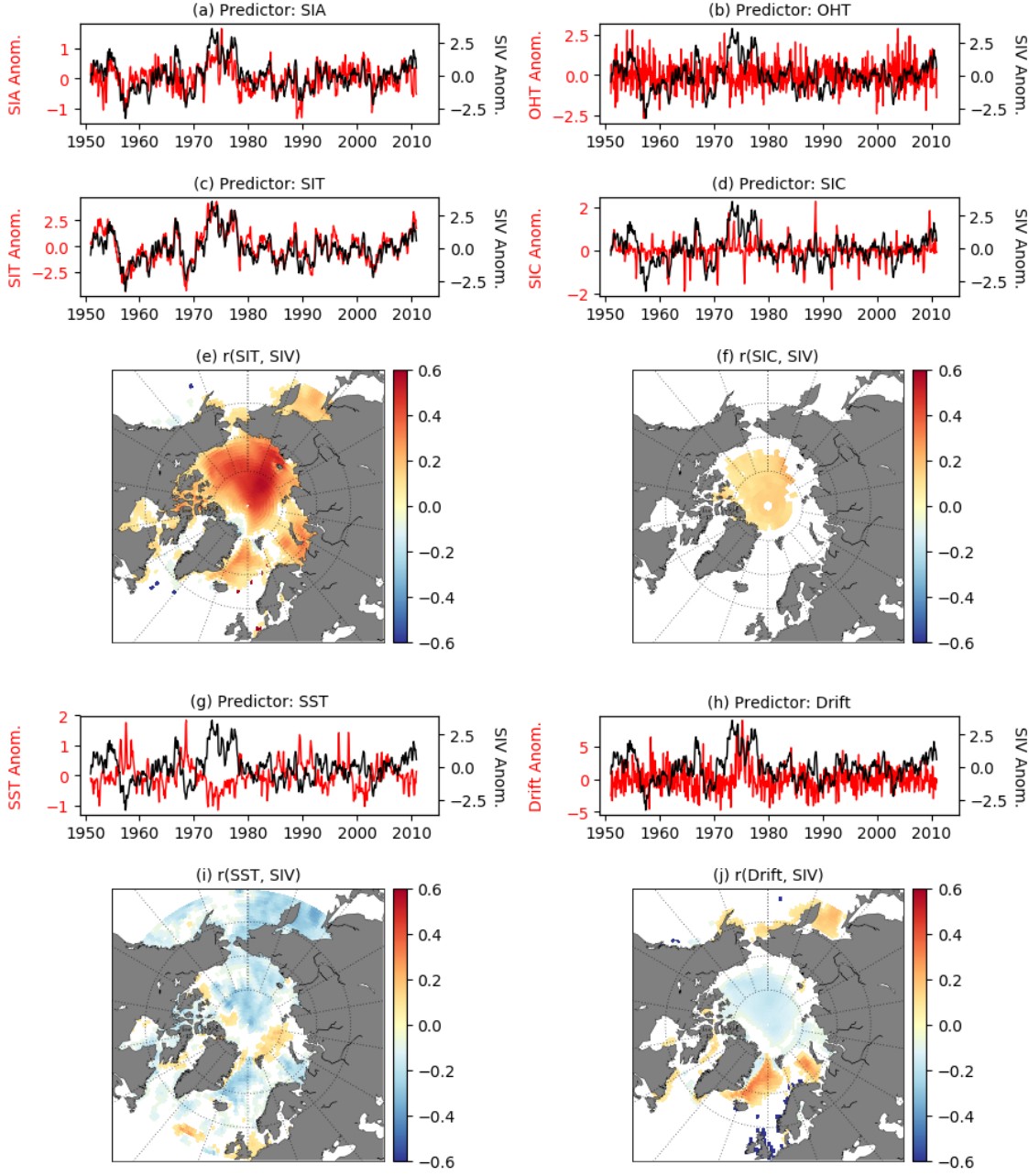

**Figure 2.** Lag-0 comparison between the time series from the predictand (SIV [$10^3$km$^3$]; black lines) and predictors: (a) SIA [$10^6$km$^2$], (b) OHT [PW], (c) SIT [m], (d) SIC [%], (g) SST [°C] and (h) Drift [km day$^{-1}$] (red lines). The correlation maps used for normalizing the regional predictors, as suggested by Drobot et al. (2006), are also shown: (e) SIT, (f) SIC, (i) SST and (j) Drift. Here, AWI-LR is merely used as an example case and not for a specific reason.

$$y = \beta_0 + \beta_1 x_1 + \beta_2 x_2 + \cdots + \beta_k x_k, \tag{1}$$

where $\beta_0$ is the constant $y$-intercept and $\beta_k$ is the slope coefficients for each explanatory variable of the empirical model.

In our case, the reconstructed time series of SIV anomaly ($SIV_{rec}$) is based on the linear relationship between this variable and the predictors aforementioned in Section 2.2. If the SIV itself is also considered as a predictor, the multiple linear regression in Eq. 1 can be written as:

$$SIV_{rec} = \beta_0 + \beta_1 SIV + \beta_2 SIA + \beta_3 OHT + \beta_4 SIT + \beta_5 SIC + \beta_6 SST + \beta_7 Drift. \tag{2}$$

To ensure robustness to the statistical reconstructions, the SEM is applied within a Monte-Carlo loop with 500 repetitions. In every repetition, 70% of the data are randomly selected for training ($N_T$) the SEM, while the remaining 30% are used for comparing ($N_C$) the original and the reconstructed SIV. In practical terms, ECMWF and HadGEM3 have 780 data points in time equivalent to the 780 months between Jan-1950 and Dec-2014 (720 for AWI; from Jan-1951 to Dec-2010) so that $N_T$ = 546 monthly values are used for building the SEM and $N_C$ = 234 values are used to evaluate how good is the SIV reconstruction ($N_T$ = 504 and $N_C$ = 216 for AWI). Since our main interest lies in the reconstruction of the SIV values, the metric used for comparing the original and reconstructed time series is the root mean squared error (RMSE). In this way, the score (Sc) of the reconstructed SIV can be represented by

$$Sc = \frac{1}{R} \sum_{r=1}^{R} \sqrt{\frac{\sum_{n=1}^{N_C} (SIV_{rec}(P) - SIV)^2}{N_C}}, \tag{3}$$

where R = 500 indicates the number of interactions in the Monte Carlo loop, $P$ represents the (set of) employed predictor(s) and the index $N_C$ emphasizes that only 30% of the data are used for comparison between original (SIV) and reconstructed SIV ($SIV_{rec}$) time series. An estimate of the Sc error ($Sc_{er}$) is given by the standard deviation calculated from the set of RMSEs given at every step of the Monte-Carlo scheme.

Two different approaches for applying the SEM are used in this work: First, in Section 3.1, we evaluate the individual and combined performances of the pan-Arctic predictors (intrinsic and averaged ones; see Section 2.2) for reconstructing the SIV anomaly at different months of the year (March and September), with a lag of one to up to 12 months upfront. Here, SIV itself is also allowed as an individual predictor to test the auto-prediction ability of this variable from lagged months. However, we are aware that SIV as a predictor could dominate the results since autocorrelation is expected to be stronger compared to the correlation with other variables. Therefore, SIV itself will not be used as a predictor in combination with other variables as generically described in Eq. 2 (see further Figs. 4h and 5h). Second, in Section 3.2, we make use of the SEM to support an optimal sampling strategy, but using the predictors in their gridded format rather than their pan-Arctic averages, as the methodology described in Section 2.4. In this case, SIV itself is not used as a predictor at all.

## 2.4 Identifying optimal sampling locations

We intend to identify a reduced number of sites from which predictor variables could offer an optimal representation of the pan-Arctic SIV anomaly. To identifying the 1st best location, a Score Map ($Sc[i,j]$) is created by applying the methodology described in Section 2.3 at each grid cell$[i,j]$. However not all grid-point predictors (SIT$[i,j]$, SIC$[i,j]$, SST$[i,j]$, Drift$[i,j]$) are necessarily used, but only the valid ones. That means, only predictors significantly correlated with the predictand are used. For instance, for the AWI-LR product, the SEM applied for a grid point placed off the eastern coast of Greenland will incorporate SIT, SST and Drift as predictors while SIC is disregarded, as suggested by the correlation maps plotted in Fig. 2e,f,i,j. SIA is the only intrinsic pan-Arctic predictor kept at this stage. The motivation for using SIA as a predictor is justified by the fact that this variable is already provided year-round by satellites so that it could be combined with *in situ* parameters in a real monitoring programme. OHT is not used at this stage since it turned out that this predictor provides a relatively poor prediction to the predictand, as discussed further in Section 3.1. Also, from an observational point of view, sampling OHT is a very complex task that requires oceanographic observations well distributed both horizontally and in depth. SIV is disregarded for an obvious reason since this is the variable that we supposedly do not have and want to predict.

By following the approach above, the goal is to create a first Score Map ($Sc[i,j]$) from which the 1st best location can be identified. In that $Sc[i,j]$ the smaller the score, the better the grid point can reproduce the pan-Arctic SIV. The 1st best location is the one represented by the smallest score in the $Sc[i,j]$. Each of the six model outputs has its first Score Map. That means that each of the datasets provides its first optimal location ($Sc[i_1,j_1]$). In practical terms, the score maps will reveal not only a single best location, but clusters of grid points defining one (or more) region(s) from where the SIV anomalies would be optimally reconstructed (see further in Section 3.2.1).

Aiming at spotting a single 1st optimal location that better represents all datasets (ensemble 1st optimal location), we take the average of the six score maps. To give the same weight for all datasets in the averaging, the individual score maps are scaled between zero and one ($ScNorm_{[i,j]}$; Calado et al. (2008)), as follows (Eq. 4):

$$ScNorm_{[i,j]} = \frac{Sc_{[i,j]} - Sc_{min}}{Sc_{max} - Sc_{min}}, \tag{4}$$

where the indexes $min$ and $max$ indicate the minimum and maximum values in the score map, respectively. Afterward, for having a coherent grid for averaging all normalized score maps, the six models are interpolated into a common $1° \times 1°$ grid. Besides the inherent different spatial grid-resolution of the models, this step has no impact on the results since the best-performing regions in the Score Maps are preserved (not shown). Finally, the 1st best ensemble sampling location is defined as the geographical coordinate where the mean ScNorm map presents its minimum value. This approach has the advantage of reducing the model dependence of the results by relying on different datasets.

After determining and fixing the 1st ideal location $[i_1,j_1]$, we search for a 2nd $[i_2,j_2]$, a 3rd $[i_3,j_3]$, and so on $[i_k,j_k]$, best locations. However, every time that a new location is identified, a region surrounding this point is also defined in order to avoid that two optimal sites are placed in close proximity of each other. To do so, we follow the concept of length scale (Blanchard-Wrigglesworth and Bitz, 2014; Ponsoni et al., 2019). The length scale defines a radius where a certain gridded variable is well-correlated to the same variable from the neighboring grid points. In this work we do not use a radius, but a

very similar approach: the correlation coefficient of our best predictor at the selected location ($SIT_{[ik,jk]}$; see Section 3.1) is calculated against the equivalent time series from all the other grid points ($SIT_{[i,j]}$). The region defined by the grid points with a correlation higher than $1/e$, a threshold for correlations below which the SIT is assumed to be uncorrelated to the point of interest, is used as a restricting region. This region is hereafter defined as "region of influence". So, all grid points enclosed into the region of influence are automatically disregarded from being selected as the next optimal location. As an example, the region of influence for a station arbitrarily placed at the North Pole, as defined by the ensemble of datasets, exhibits departures from concentric reflecting the transpolar drift (Fig. 3).

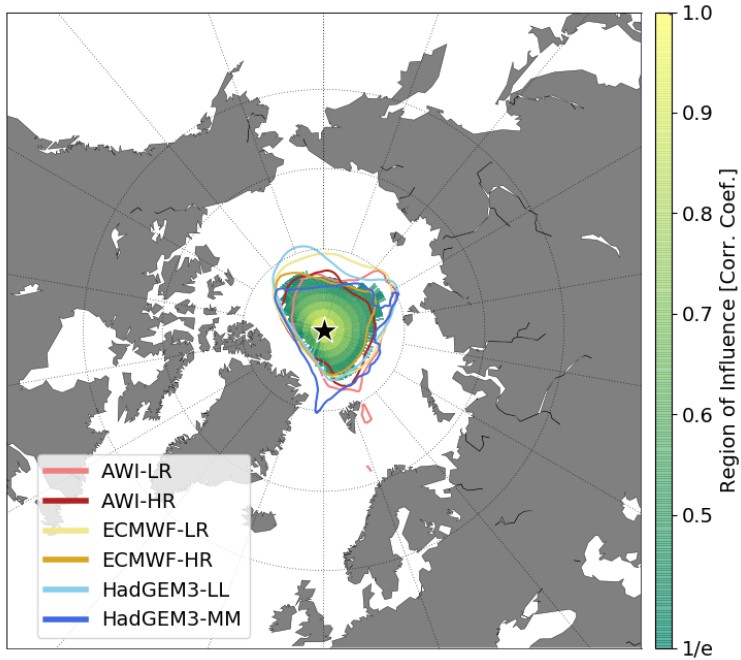

**Figure 3.** Region of influence for a station arbitrarily placed at the North Pole (black star) as defined by each model (colorful lines) and by the averaged region of influence from the different models (shades of green to yellow).

In this approach, the regression described in Eq. 2, with $k$ optimal locations, takes the following format:

$$SIV_{rec} = \beta_0 + \beta_2 SIA + \sum \beta_{P1[i_1,j_1]} P1[i_1,j_1] + \sum \beta_{P2[i_2,j_2]} P2[i_2,j_2] + \cdots + \sum \beta_{Pk[i_k,j_k]} Pk[i_k,j_k], \tag{5}$$

where the term $\beta_{Pk[i_k,j_k]} Pk[i_k,j_k]$ represents the product between the valid predictors $Pk[i_k,j_k]$, at the optimal location number $k$, and their respective slope coefficients $\beta_{Pk[i_k,j_k]}$. It is worthwhile mentioning that only valid predictors, which means only predictors from grid points placed outside the region of influence defined by previously selected points, and that are validated by the correlation map criterion, are used in Eq. 5.

## 3 Results

### 3.1 Statistical predictability of SIV anomaly: pan-Arctic predictors

In this section, the statistical predictability of the SIV anomaly is quantitatively evaluated by considering leading periods of one to 12 months. Also, the predictive performance of seven pan-Arctic predictors is tested. The predictors are SIV itself,

SIA, OHT, SIT, SIC, SST and Drift. Here, we focus on the months with relatively large (March; Section 3.1.1) and reduced (September; Section 3.1.2) SIV at the end of the winter and summer, respectively.

### 3.1.1 Statistical predictability of March SIV anomaly: pan-Arctic predictors

Figure 4 displays the predictive performance (quantified by the RMSE) of different predictors for estimating March SIV anomalies. The SIV itself is the best predictor variable and its score gradually increases from 12 (Sc = 1.0 $\times 10^3$km$^3$) to four

(Sc = 0.68 $\times 10^3$km$^3$) leading months. During this period the mean performance for the ensemble of models increases by about 32%. As per three leading months, from December to February, the predictive capability substantially improves by 43% (Sc = 0.57 $\times 10^3$km$^3$), 59% (Sc = 0.41 $\times 10^3$km$^3$) and 77% (Sc = 0.23 $\times 10^3$km$^3$), respectively (Fig. 4a). The second best predictor is the SIT, which has performance similar to the SIV predictor from about 12 to nine leading months (ensemble mean Sc = 1.02 $\times 10^3$km$^3$, 1.03 $\times 10^3$km$^3$, 1.0 $\times 10^3$km$^3$; Fig. 4d). Nevertheless, its score remains relatively stable and improves

only by about 25%, from May to February (Sc = 1.0 and 0.75 $\times 10^3$km$^3$). SIC (Fig. 4e), SST (Fig. 4f) and Drift (Fig. 4g) have poorer performance compared to SIT, but similar behavior with the score slightly improving over time until one leading month. SIA (Fig. 4b) is a valid predictor for AWI and HadGEM3 models, but it does not seem to be the case for ECMWF versions. Finally, OHT showed to be a poor predictor in terms of monthly predictability. For most of the leading months and models, the statistical reconstruction is not significant when estimated with this predictor (Fig. 4c).

A way of further improving the statistical predictability is to use several predictors at once. Figure 4h shows the case where all the aforementioned predictors (except SIV) are used by the empirical model. For this configuration, the predictive skill is still 10% lower than the case where SIV is standing alone as a predictor, but it is about 10% better than the reconstructions provided only by the SIT. The inter-model comparison does not show a conclusive answer to the question of whether or not the model resolution plays a role in the statistical predictability of March SIV anomalies. Overall, AWI-HR predictors are more

skilled than AWI-LR predictors, though the opposite is observed for HadGEM3. For the ECMWF versions, the SIV anomalies from EMCWF-HR present better reproducibility, while ECMWF-LR presents much larger errors. Note that ECMWF-LR has a mean state characterized by a much thicker sea ice and, consequently, higher variance (see Fig. 1). This is the reason that makes ECMWF-LR an outlier compared to the other five model outputs for this and other results found in this manuscript (see further discussion in Section 4).

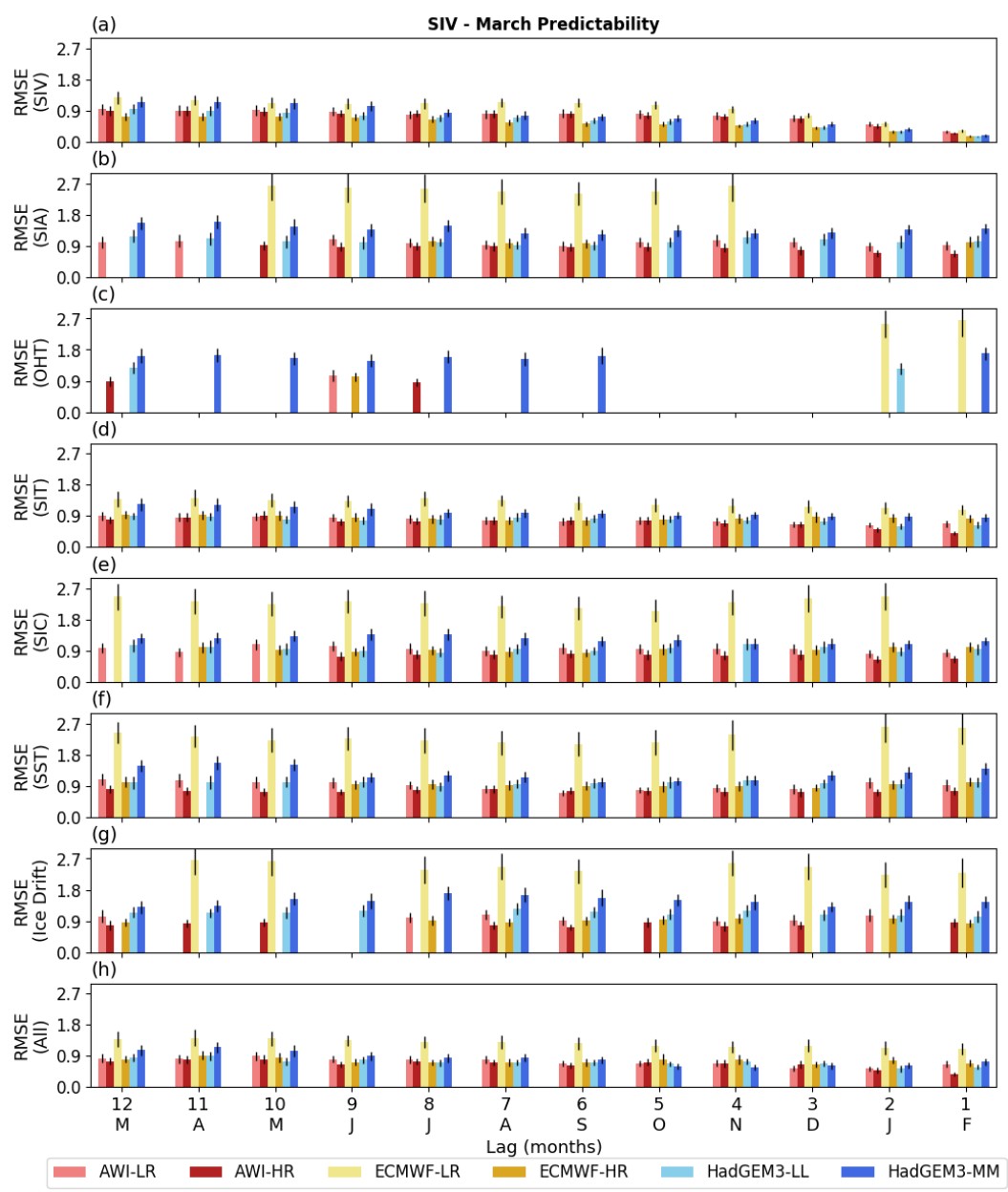

**Figure 4.** Statistical predictability of the March SIV anomalies, estimated from 12-leading months and quantified by the RMSE ($10^3$km$^3$) calculated between the original and reconstructed time series (Sc), as prescribed by seven predictors: (a) SIV itself, (b) SIA, (c) OHT, (d) SIT, (e) SIC, (f) SST, (g) Drift. The predictions employing all predictor variables (except the SIV itself) are displayed in (h). The vertical black lines indicate the error as provided by the 500 Monte Carlo simulations. The statistical predictability follows the methodology introduced in Section 2.3. Missing vertical bars mean that the statistical reconstruction is not statistically significant. The long-term trend and seasonal cycle are excluded from both predictand and predictors.

### 3.1.2 Statistical predictability of September SIV anomaly: pan-Arctic predictors

A similar scenario compared to March is found for the September SIV anomaly predictability (Fig. 5). The best predictor is the SIV itself (Fig. 5a) for which the predictive skill improves by about 83.6% from previous September to August (Sc $= 1.16 \times 10^3 \text{km}^3$ and $0.19 \times 10^3 \text{km}^3$). This improvement is mainly attributed to the three months before September: Sc $= 0.71 \times 10^3 \text{km}^3$, $0.44 \times 10^3 \text{km}^3$ and $0.19 \times 10^3 \text{km}^3$ for June, July and August, respectively. The second best predictor is SIT (Fig. 5d), while SIC (Fig. 5e), SST (Fig. 5f) and Drift (Fig. 5g) present an intermediate performance. For the former four predictors, the ensemble mean Sc slightly improves from 12 to one leading months in about: 28.8% (Sc $= 1.04 \times 10^3 \text{km}^3$ and $0.74 \times 10^3 \text{km}^3$), 15% (Sc $= 1.40 \times 10^3 \text{km}^3$ and $1.19 \times 10^3 \text{km}^3$), 29% (Sc $= 1.26 \times 10^3 \text{km}^3$ and $0.90 \times 10^3 \text{km}^3$) and 24% (Sc $= 1.46 \times 10^3 \text{km}^3$ and $1.11 \times 10^3 \text{km}^3$), respectively. Not all tested predictors are statistically significant for reproducing the SIV anomalies. Again, this is the case for OHT (Fig. 5c). SIA also presents poor performance for some models and leading months (Fig. 5b). Another resemblance to March predictability is the relatively poor performance presented by the predictor variables from ECMWF-LR.

## 3.2 Statistical predictability of SIV anomaly: regional predictors

In this section, the empirical statistical model is used for supporting an optimal sampling strategy by following the methodology described in Section 2.4. To do so, we evaluate the predictors at every grid-point rather than use their pan-Arctic averages. The reasoning behind this approach lies in the hypothesis that the statistical empirical model can fairly reproduce and/or predict the SIV anomalies if a few optimal locations provide *in situ* measurements from the predictor variables. These *in situ* observations can be applied concomitantly with predictors that are continuously measured by satellites as the pan-Arctic SIA and the SIC.

Here we assume that numerical models are able to reproduce the main physical processes behind the interactions among predictand and predictors. Practically, we will take into account four gridded predictors that are SIT, SIC, SST and Drift, and one pan-Arctic predictor that is SIA, although it is worthwhile reminding ourselves that only predictors significantly correlated with the predictand will be incorporated to the statistical model. As per the results of Section 3.1, the OHT will not be included as predictor variable due to its poor capability to provide a skillful prediction, being reinforced by the difficulties associated with the *in situ* sampling and estimation of this variable.

### 3.2.1 Optimal sampling locations

For each of the six model realizations, score maps (Sc[$i,j$]; Eq. 3) were determined with the aim of spotting the location that can better reproduce the SIV anomalies as shown in Fig. 6. This location is so defined as the grid point with minimum RMSE calculated between the original and reconstructed time series (Sc[$i1,j1$]; black stars in Fig. 6). The resulting ideal location for AWI-LR, AWI-HR, and HadGEM-LL (Fig. 6a,b,e) are relatively close to each other, separated by a maximum of ∼600 km. Even though ECMWF-LR, ECMWF-HR, and HadGEM3-MM (Fig. 6c,d,f) suggest optimal locations that are placed farther from the sites suggested by the other datasets, their score maps still suggest a relatively good skill (low RMSE values) at the

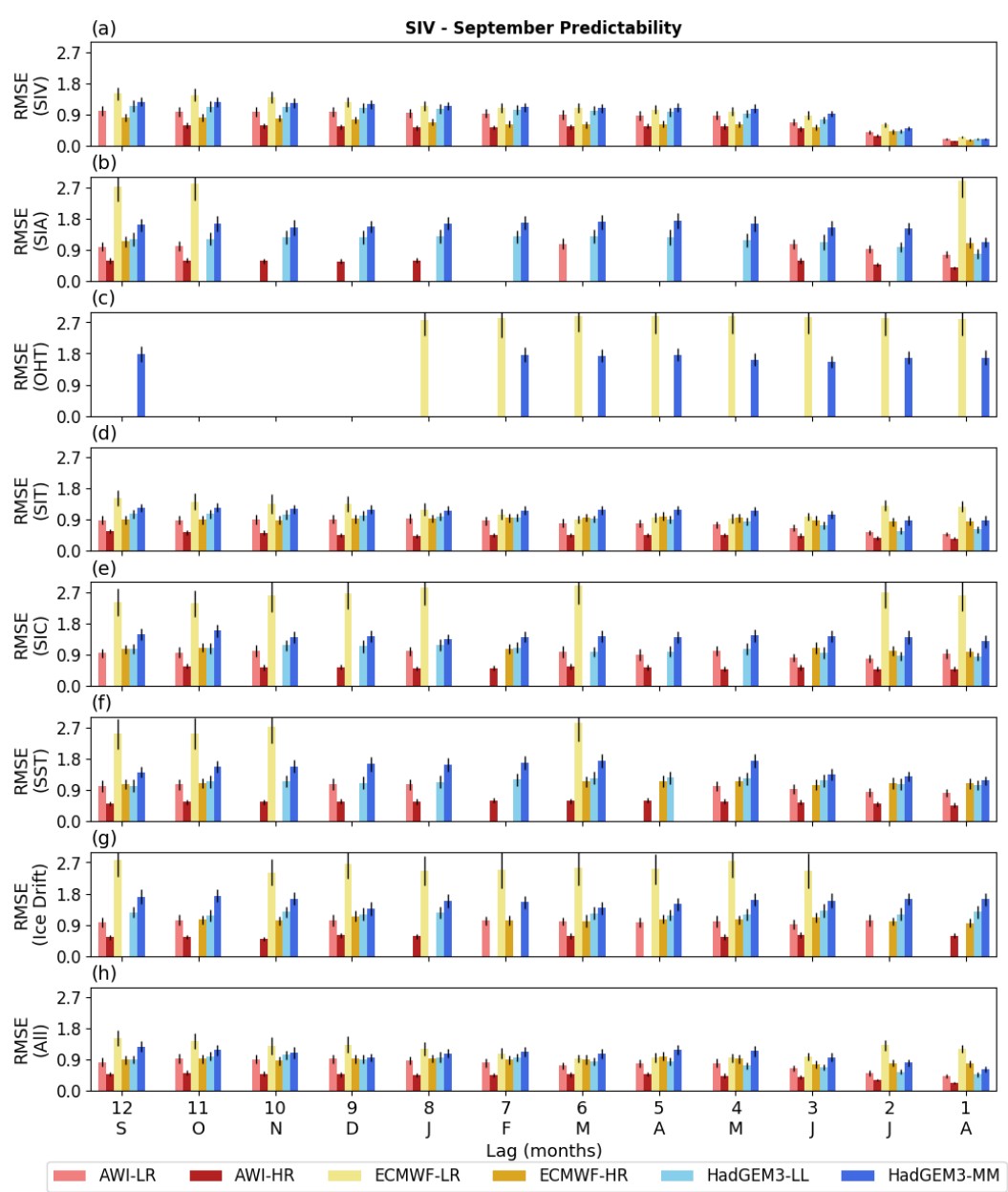

**Figure 5.** Statistical predictability of the September SIV anomalies, estimated from 12-leading months and quantified by the RMSE ($10^3$km$^3$) calculated between the original and reconstructed time series (Sc), as prescribed by seven predictors: (a) SIV itself, (b) SIA, (c) OHT, (d) SIT, (e) SIC, (f) SST, (g) Drift. The predictions employing all predictor variables (except the SIV itself) are displayed in (h). The vertical black lines indicate the error as provided by the 500 Monte Carlo simulations. The statistical predictability follows the methodology introduced in Section 2.3. Missing vertical bars mean that the statistical reconstruction is not statistically significant. The long-term trend and seasonal cycle are excluded from both predictand and predictors.

common region occupied by the three previous referred models. This fact justifies further the multi-model approach used in this work.

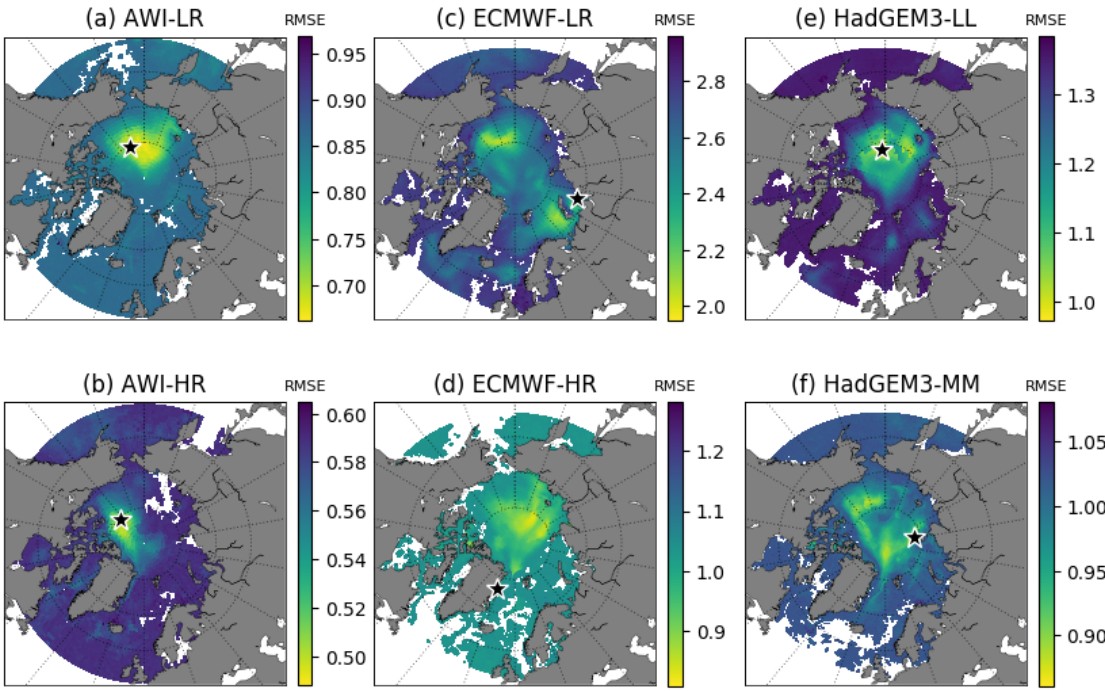

**Figure 6.** Score Maps (Sc[$i,j$]) represented by the RMSE ($10^3$km$^3$) calculated at every grid cell between the original and the reconstructed SIV anomalies. The smaller the RMSE error (shades of yellow), the higher the performance of the grid point for reconstructing the SIV anomaly. The black star indicates the primary optimal location for each model Sc[$i1,j1$]. Note that the colormap scale is different for each map.

The RMSEs (and associated STD from the Monte-Carlo scheme) calculated between the original SIV anomalies and the SIV anomalies reconstructed by the SEM, using predictor variables from the 1st optimal location (black stars in Fig. 6), are shown in the mid column of Table 2. Based on those values, predictor variables from the AWI systems can better reproduce the SIV anomalies compared to the predictors from HadGEM and ECMWF. For the three models, the high-resolution version provides better statistical predictability. A common score map, with the indication of a common 1st optimal location placed at the transition Chukchi Sea – Central Arctic – Beaufort Sea (158.0°W, 79.5°N), is shown in Fig. 7a. This common location is found through the ensemble mean of the scaled individual score maps, following the methodology described in Section 2.4. The RMSE values from that common location (Fig. 7a), but retrieved from the score maps in Fig. 6, are shown in the right column of Table 2. The predictive skill drops by about 10% when the common point is chosen for all models, except for AWI-LR which presents similar results for the two locations. These values reinforce that, at least for this 1st location, the predictors from the

**Table 2.** Mean RMSEs (and associated STDs) from the 500-Monte-Carlo realizations calculated between the original SIV anomalies and the SIV anomalies reconstructed by the SEM. We recall that in each Monte-Carlo realization 70% of the data is randomly used for training the SEM, while 30% is used for calculating the error. The mid column shows the values for the case where the predictors are extracted from the individual optimal locations, while the right column shows the values found with predictors from the common optimal location.

| Models | RMSE (Error) $\times 10^3$km$^3$ 1st Optimal Location Individual location | RMSE (STD) $\times 10^3$km$^3$ 1st Optimal Location Common location |
|---|---|---|
| AWI-LR | 0.66 ($\pm$0.03) | 0.67 ($\pm$0.03) |
| AWI-HR | 0.49 ($\pm$0.02) | 0.54 ($\pm$0.02) |
| ECMWF-LR | 1.95 ($\pm$0.06) | 2.11 ($\pm$0.09) |
| ECMWF-HR | 0.81 ($\pm$0.03) | 0.91 ($\pm$0.04) |
| HadGEM3-LL | 0.97 ($\pm$0.04) | 1.09 ($\pm$0.05) |
| HadGEM3-MM | 0.86 ($\pm$0.05) | 0.95 ($\pm$0.04) |

high-resolution outputs lead to a better predictive skill compared to the low-resolution predictors from their counterpart. Note that this was not the case when using pan-Arctic predictors in Section 3.1.

Once the primary common optimal site has been identified and accepted for all datasets, we search for the 2nd best location. For that, the neighboring grid points which fell into the region of influence of the 1st best site are not considered as a second option. Fig. 7b shows the 1st location's region of influence. The procedure followed for identifying the 1st and 2nd sites is so repeated for the $n$th next locations. Aiming at improving the reconstruction of the SIV anomalies, every time that a new location is set, the valid predictors from this new point add to the predictors from the previous stations into the SEM. Fig. 7c,e,g,i show the 2nd to the 5th optimal sites accompanied by their respective regions of influence (Fig. 7d,f,h,j). The 2nd site is about 167 km from the North Pole. The 3rd, 4th and 5th points are placed at the offshore domain of the Laptev Sea near the transition with the Central Arctic, in the Central Arctic to the north of the Canadian Islands, and within the Beaufort Sea, respectively.

Fig. 8 represents an idealized scenario with the ten best locations and their respective regions of influence. In such a context, the selection of points respects the hierarchy of the regions of influence in a way that the 2nd site can not be placed within the region of influence #1 (shades of red), the 3rd point can not be placed within the regions of influence #1 and #2 (shades of red and purple), and so on. Note that with the proposed methodology, the regions of influence from the ten first locations are covering almost the entire Arctic Ocean and adjacent seas, with exception of the Canadian Archipelago, the Kara Sea, and the Greenland Sea (see Fig. 9). But even for the two later cases, the region of influence from other locations are partially covering these seas (Fig. 9; black line). The question of whether or not is indeed required all ten locations to fairly predict the SIV anomalies, both in terms of anomaly values and variability, will be answered in the next sections.

Table 3 displays the geographical coordinates of the ten locations as well as the Arctic sub-regions occupied by them, as identified in Fig. 9. The division of the Arctic in sub-regions is based on the classical definition adopted by the broadly used Multisensor Analyzed Sea Ice Extent - Northern Hemisphere (MASIE-NH) product, which is made available by the National

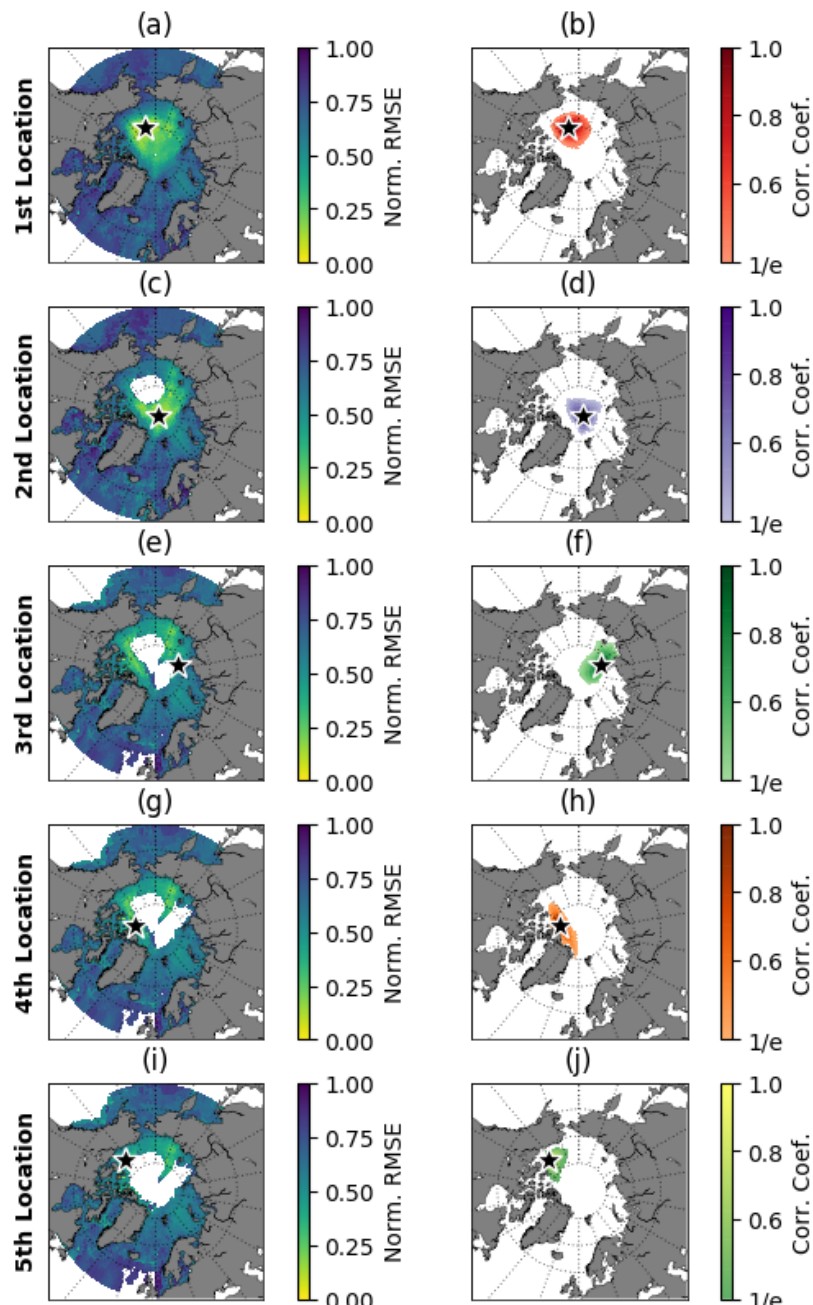

**Figure 7.** (a) Ensemble mean–normalized score map (ScNorm) for the 1st best sampling location. (b) The region of influence is defined for the 1st best location. The panels (c,d), (e,f), (g,h) and (i,j) represent the same as (a,b) but for the 2nd, 3rd, 4th and 5th best sampling locations, respectively.

Snow & Ice Data Center (NSIDC). Most of the stations are placed within the Central Arctic (2nd, 4th, and 8th) or, as mentioned above, in the transition of this region with the Chukchi Sea (1st) and Laptev Sea (3rd), where the sea ice tends to be perennial. The 5th location is placed at the central part of the Beaufort Sea. The 6th and 9th stations are located at the offshore and inshore limits of the East Siberian Sea, respectively. The 7th site is suggested to be at the Barents Sea off the Severny Island and the 10th station is occupying the near-coastal Laptev Sea.

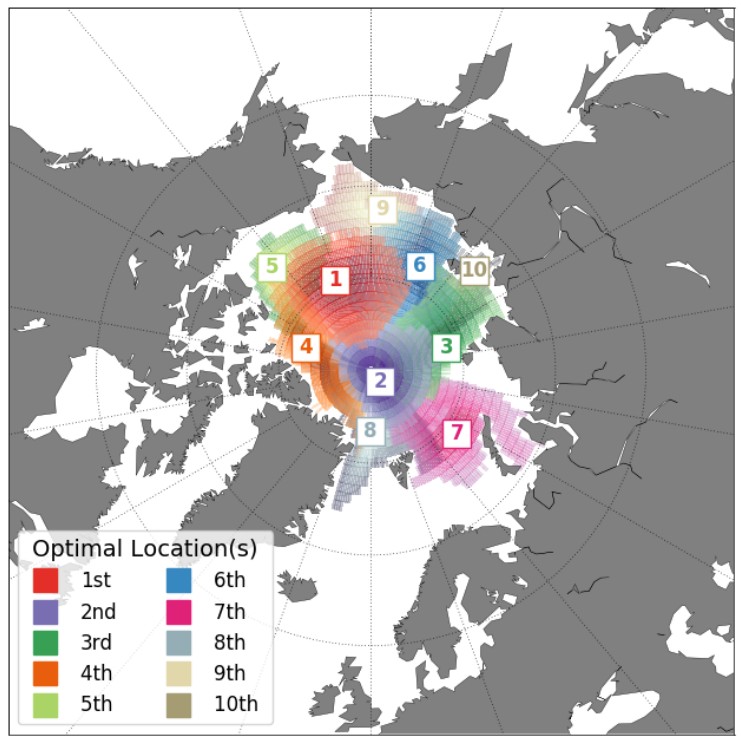

**Figure 8.** Optimal observing framework, as suggested by the ensemble of model outputs, for sampling predictor variables in order to statistically reconstruct and/or predict the Pan-Arctic SIV anomaly. The numbers indicate the 1st up to the 10th best observing locations in the respective order. The hatched area around each location (same color code) represents their respective region of influence. The selection of points respects the hierarchy of the regions of influence in a way that the 2nd point can not be placed within the region of influence #1 (shades of red), the 3rd point can not be placed within the regions of influence #1 and #2 (shades of red and purple), and so on.

### 3.2.2 Reconstructed SIV anomaly

Once the ideal sampling locations are established, these sites are used to effectively reconstruct the entire time series of SIV anomalies from the six model outputs, by taking into account only the valid predictors from each location. We will make use of the RMSE to evaluate how good is our statistical prediction in terms of absolute values as in the previous sections, but here we are also interested in inspecting the ability of the empirical model to reproduce the full variability of the SIV anomalies.

**Table 3.** Geographical coordinates for the first ten optimal sampling locations (second and third columns). The fourth column informs the sub-regions in which each of the points are placed in (see Fig. 9). The limits of the sub-regions are suggested by the National Snow & Ice Data Center (NSIDC).

| Optimal Location | Latitude | Longitude | Sub-Region |
|:---:|:---:|:---:|:---|
| #1 | 79.5°N | 158.0°W | Chukchi Sea (CS) |
| #2 | 88.5°N | 040.0°E | Central Arctic (CA) |
| #3 | 81.5°N | 107.0°E | Laptev Sea (LS) |
| #4 | 82.5°N | 109.0°W | Central Arctic (CA) |
| #5 | 74.5°N | 136.0°W | Beaufort Sea (BeS) |
| #6 | 77.5°N | 155.0°E | East Siberian Sea (ESS) |
| #7 | 78.5°N | 054.0°E | Barents Sea (BrS) |
| #8 | 83.5°N | 001.0°W | Central Arctic (CA) |
| #9 | 72.5°N | 176.0°E | East Siberian Sea (ESS) |
| #10 | 74.5°N | 134.0°E | Laptev Sea (LS) |

For that, apart from the RMSE, we also calculate the coefficient of determination ($R^2$) between the original and reconstructed time series.

Figure 10 provides a comparison at lag-0 between the original (black lines) and the reconstructed times series by taking into account the 1st (red lines), the three first (green lines) and the six first (blue lines) locations. For the first reconstruction, RMSE
values are almost identical to the ones shown in the second column of Table 2 (see Fig. 11a; $y$-axis=1). Again, for all three models, the predictor variables from the higher resolution versions present better performance in reproducing the SIV anomaly values. The relatively poor skill of the ECMWF-LR predictors compared to the other five systems is remarkable (Fig. 10c).

Figure 11a summarizes the RMSE values for the reconstructions conducted with data from only the 1st up to all ten combined locations. The pattern of better prediction skill for the models with higher grid resolution revealed by the 1st location remains
when more sites are incorporated into the SEM. From the ensemble means the RMSE ($\times 10^3 km^3$) values are, respectively, 1.06, 0.95, 0.90, 0.81, 0.78, 0.70, 0.65, 0.63, 0.60 and 0.59 for the reconstruction with one to ten locations (black curve/points in Fig. 11a). By excluding the outliers from ECMWF-LR, the previous RMSEs reduce to about 20% as shown by the gray curve-points in Fig. 11a). For most of the datasets, the statistical reconstruction seems to improve better until the incorporation of the 5th to 6th locations, from when on the improvement seems to attenuate (Fig. 11a).

Figure 11b introduces a similar analysis but quantified by the $R^2$. Interestingly, for this metric, the ECMWF-LR is not outstanding from the others, and its predictors present a similar performance for reproducing the SIV anomaly variability. By account the reconstructions with one to ten optimal sites, the ensemble means of $R^2$ values are: 0.53, 0.63, 0.67, 0.73, 0.75, 0.80, 0.81, 0.83, 0.84 and 0.84, respectively. These ensemble means suggest that the statistical empirical model could reproduce more than 60% of the SIV variability by using predictors from only the three first optimal locations. AWI and HadGEM datasets
indicate that four locations are sufficient for reproducing more than 70% of the variability. With six well-positioned sites, about

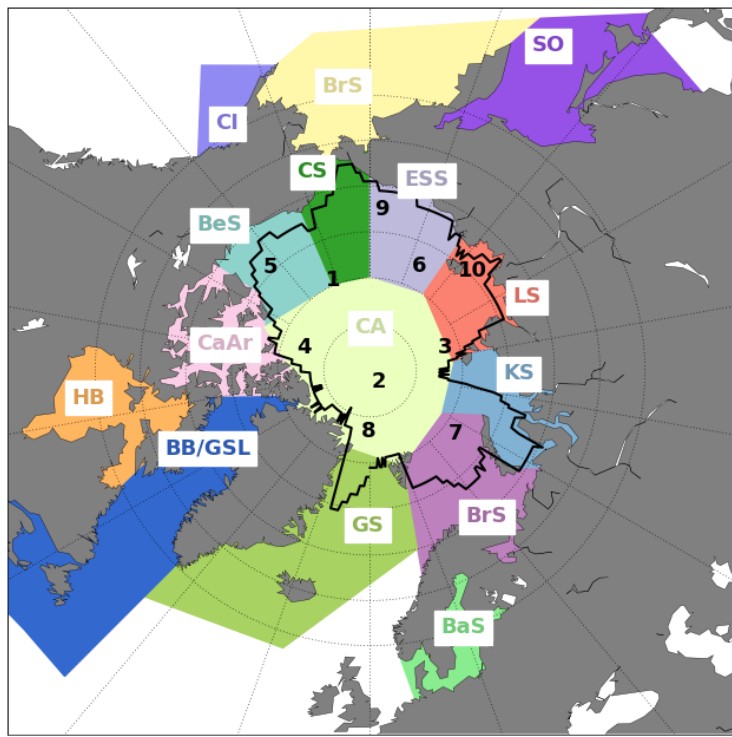

**Figure 9.** Optimal observing framework for sampling predictor variables in order to statistically reconstruct and/or predict the Pan-Arctic SIV anomaly. The numbers indicate the 1st up to the 10th optimal sites. Each of the colored areas represent an Arctic sub-region according to the Arctic subdivision suggested by the National Snow & Ice Data Center (NSIDC). The black line indicates the global region of influence defined in Fig. 8 (color-shaded areas). Acronyms: Beaufort Sea (BeS); Chukchi Sea (CS); East Siberian Sea (ESS); Laptev Sea (LS); Kara Sea (KS); Barents Sea (BrS); Greenland Sea (GS); Baffin Bay/Gulf of St. Lawrence (BeS); Canadian Archipelago (CaAr); Hudson Bay (HB); Central Arctic (CA); Bering Sea (BrS); Baltic Sea (BaS); Sea of Okhotsk (SO); Cook Inlet (CI).

80% of the SIV anomaly may be explained as suggested by the ensemble mean (Fig. 11b). As per the 6th station, the gain from adding new locations seems to be minimal ($\sim$1%). In addition, it is of interest that the $R^2$ metric behaves in opposition to the RMSE since the best performing predictors are the ones coming from the model's version with lower grid resolution.

In terms of used predictor variables, Fig. 11c reiterates that SIT is the most skillful of the predictors. From the 60 cases that the SEM was applied (six datasets, ten locations), SIT was used 59 times. SIT was not a valid predictor only for the 9th location in ECMWF-HR. SST and Drift were used in about two thirds (40 and 38 times, respectively), while SIC was used only in half (29 times) of the cases. Inspecting the individual model outputs, HadGEM (the two resolutions comprised) is the one in which the empirical model takes the best advantage of the available gridded predictors, having neglected one of them in only 15 out of 80 cases, while ECMWF and AWI have ignored predictors in 29 and 30 out of 80 cases, respectively.

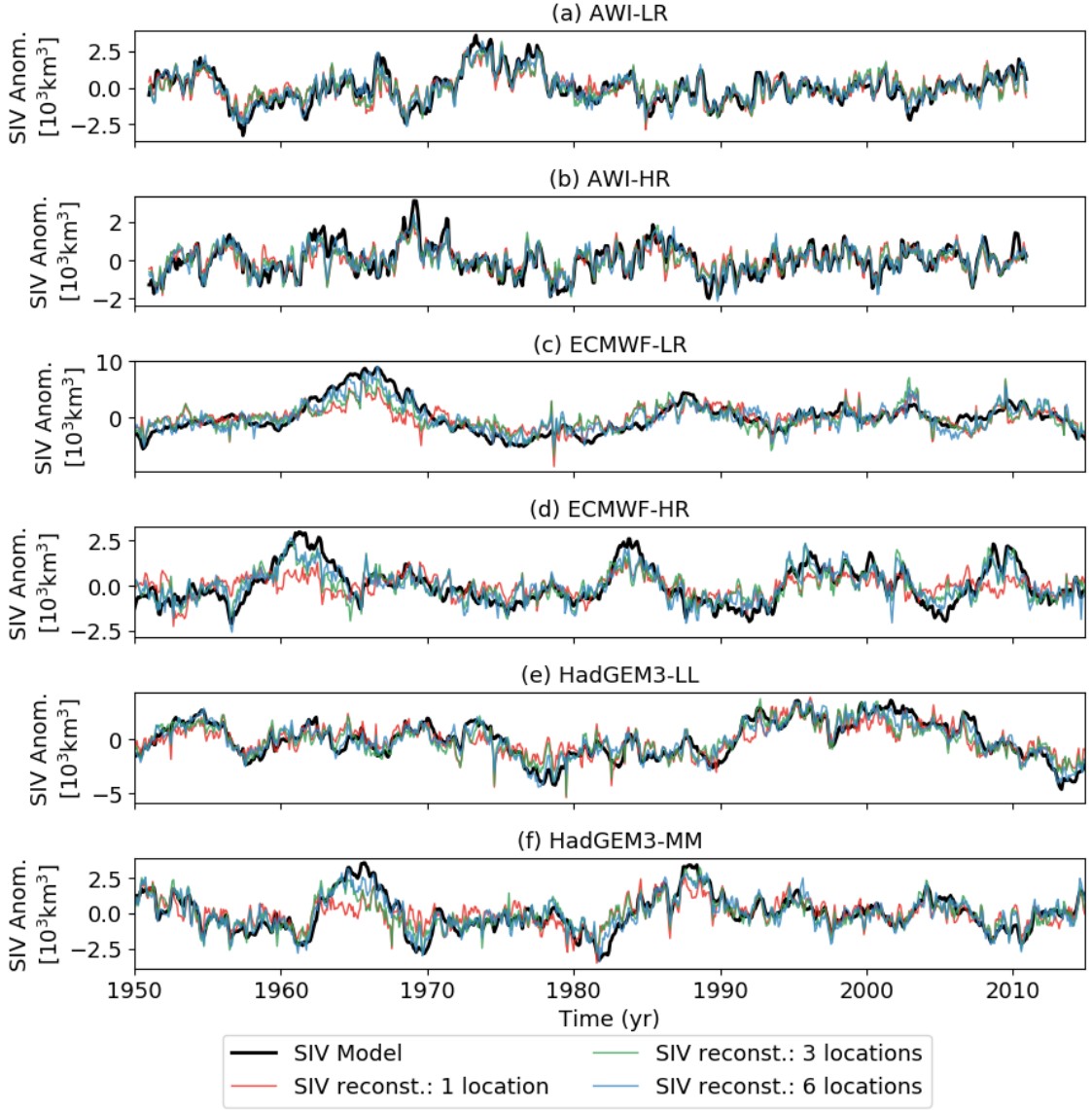

**Figure 10.** Lag-0 comparision between the original (black) and statistically reconstructed SIV anomalies. The reconstruction takes into account the 1st (red), the three first (1st–3rd; green) and the six first (1st–6th; blue) optimal locations: (a) AWI-LR, (b) AWI-HR, (c) ECMWF-LR, (d) ECMWF-HR, (e) HadGEM-LL and (f) HadGEM-MM. Note the different scales in the y-axes.

To evaluate the performance and robustness of our SEM, the RMSE and $R^2$ calculated between the original and our-methodology-based reconstructed SIV anomalies (Fig. 11a,b) are compared against the same two metrics but now estimated by a simple multiple linear regression model having as input predictor data from randomly chosen locations (Fig. 12). For that purpose, 100 combinations of ten randomly chosen locations were determined. For each combination, the SIV anomaly is

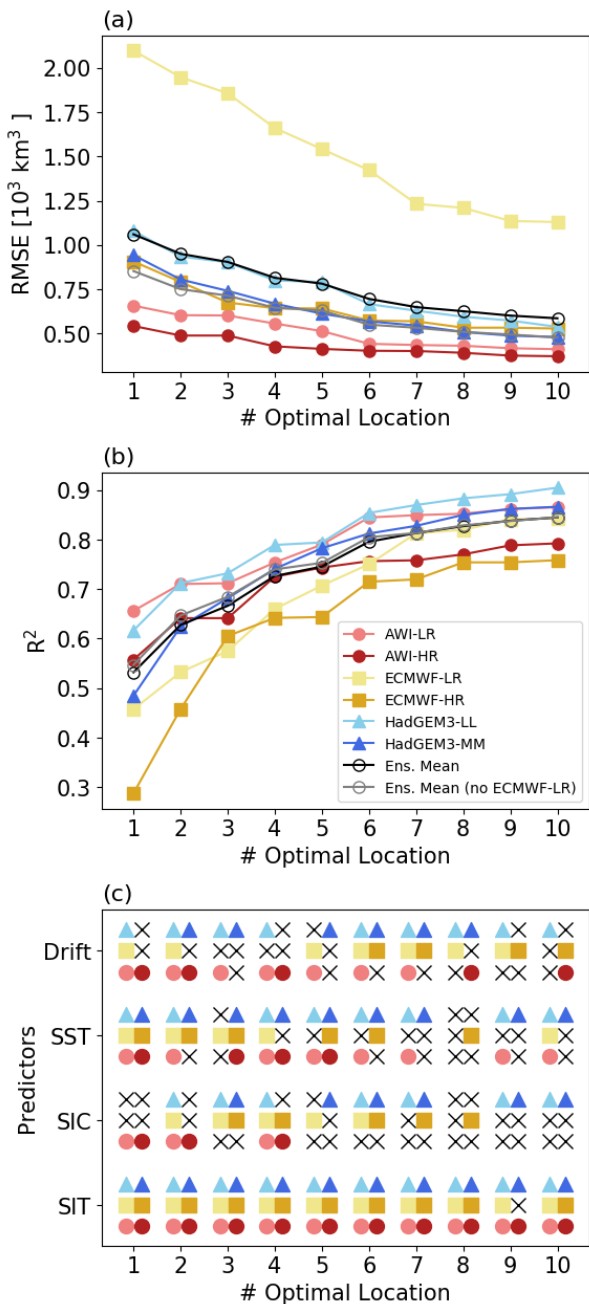

**Figure 11.** (a) RMSE (y-axis) estimated between the original and reconstructed time series by taking into account predictor variables from one up to ten optimally selected locations (x-axis). (b) Same as (a) but using $R^2$ (y-axis) to compare original and reconstructed time series. (c) Valid predictors, as determined by the correlation maps, retrieved from each targeted location. If a predictor is valid (y-axis), its respective symbol, as defined in the inset legend from (b), is plotted. A black cross indicates that the predictor is not valid at the respective location.

reconstructed with predictor data from the 1st location, the 1st–2nd, the 1st–3rd, ..., the 1st–10th locations. We have used the same predictor variables from randomly locations placed only into the global region of influence represented by the black line in Fig. 9. The results show that the SIV reconstructions based on our methodology (and optimally selected locations) are more skillful both in terms of RMSE and $R^2$. This is valid for all models, considering a single location or any combination of up to ten locations (Fig. 12).

## 4 Discussion

In this work, we have introduced a statistical empirical model for predicting the Arctic SIV anomaly on the interannual time scale. The model was built and tested with data from three AOGCMs (AWI-CM, ECMWF-IFS, and HadGEM3-GC3.1), each of which provided with two horizontal resolutions, performing a total of six datasets. We have first inspected the predictive skill of seven different pan-Arctic predictors, namely: SIV, SIA, OHT, SIT, SIC, SST, and Drift. These predictors were tested since they have dynamical and/or thermodynamical influence on the SIV. The three first are intrinsically represented by single time series, while the remaining are gridded variables that were reduced to mean pan-Arctic time series. From this first assessment, performed for the months of March and September, the results (Section 3.1) show that the best predictors are the SIV itself and the SIT, whilst SST, Drift, SIC and SIA provide some intermediate-skill predictions. In general, such results are valid for predictions performed from one back to 12 leading months. For the SIV predictor, the skill substantially increases in the last three leading months. For the remaining aforementioned predictors, the skill slightly improves from 12 to one leading month.

In contrast, OHT provided very poor predictive skill. Docquier et al. (2019) recently showed (their Fig. 12) a relatively good correlation between OHT and the SIV. However, these authors correlated annual averages of OHT against monthly values of SIV, but here we are considering monthly means for all predictors. Based on that, the results from both manuscripts suggest that the OHT has a cumulative impact on the sea ice throughout the year, which is not so remarkable when looking at individual months, even if several leading months are considered. One might wonder how SST is a relatively skillful predictor, while OHT not. We reiterate that the OHT tested as a predictor in this study is a remote parameter, which takes into account the seawater temperature (and meridional velocities) throughout the entire water column, calculated at 60.0°N for the Atlantic basin ocean (Docquier et al., 2019). There are other potential candidates to explain why OHT is a poor predictor, as for instance model biases such as an overestimation of the stratification at the near-surface layer, which could attenuate the heat content being transported towards the Arctic Ocean. Nevertheless, this is a subject that requires a more detailed investigation. From Section 3.1's results is also noticeable that the ECMWF-LR predictors present a relatively poor skill compared to the others. This is explained by the fact that this model has a mean state characterized by a much thicker sea ice (see Fig. 1), impacting the RMSE used as a metric for evaluating the prediction skill.

We now recall and objectively answer the first open question posed in the introduction of this manuscript:

**(i) What is the performance of different pan-Arctic predictors for predicting pan-Arctic SIV anomalies?**
Taking into account the ensemble mean, and using the average RMSE calculated between original and reconstructed SIV time series (Section 3.1; Figs. 4 and 5) for the last three leading months as score, the best predictors for March are sorted in the

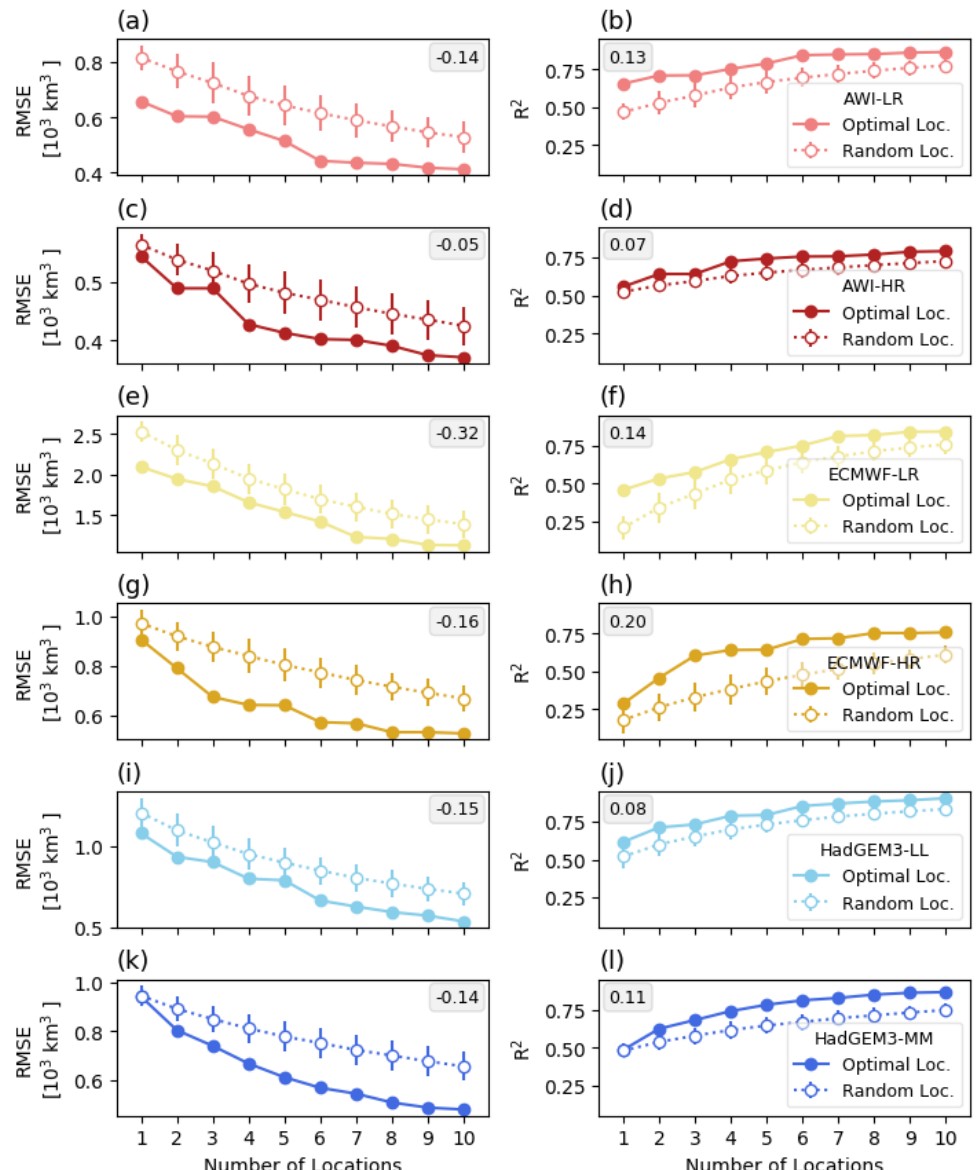

**Figure 12.** Root Mean Squared Error (RMSE; left column) and coefficient of determination ($R^2$; right column) calculated between the original and reconstructed SIV anomalies. The reconstructed SIV volume anomalies are based on the optimally selected locations following our methodology (full dots; same as in Fig. 11a,b), as well as by randomly chosen locations (empty dots). In the last case, 100 sets of ten randomly chosen locations are used. For each of the 100 sets, the SIV anomaly is reconstructed using data from the 1st, the 1st–2nd, the 1st–3rd, ..., the 1st–10th, locations. The random locations are all enclosed into the global region of influence defined (Fig. 9; black line). The vertical bars associated with the empty dots represent the one standard deviation from the 100 reconstructions. The inset numbers represent the average difference between the two curves shown in each sub-panel.

following order: SIV ($0.41 \times 10^3$km$^3$), SIT ($0.78 \times 10^3$km$^3$), SIA ($1.01 \times 10^3$km$^3$), SIC ($1.10 \times 10^3$km$^3$), SST ($1.15 \times 10^3$km$^3$), Drift ($1.32 \times 10^3$km$^3$) and OHT ($2.05 \times 10^3$km$^3$). The best predictors for September are sorted as: SIV ($0.45 \times 10^3$km$^3$), SIT ($0.76 \times 10^3$km$^3$), SST ($0.96 \times 10^3$km$^3$), SIA ($1.07 \times 10^3$km$^3$), SIC ($1.12 \times 10^3$km$^3$), Drift ($1.22 \times 10^3$km$^3$) and OHT ($2.24 \times 10^3$ km$^3$). If all predictors are used (except SIV itself), the averaged scores for three leading months are $0.70 \times 10^3$km$^3$ for both March and September.

Once the statistical empirical model had been developed and the potential predictor variables are identified, we can make use of this information for recommending an optimal observing system. For example, such a system could be part of an operational oceanography programme in which predictor data are provided to the statistical model through in situ observations (e.g., oceanographic moorings and/or buoys) of SIT, SST, and Drift. The SIC and pan-Arctic SIA could also be incorporated into the statistical model since they are regularly sampled by satellites. OHT and the SIV are disregarded as predictors. The former is not a skillful predictor (as shown in Section 3.1), while the latter is the variable that we want to predict. We restricted our analyses to a maximum of ten optimal locations, although a reduced number of observational sites are sufficient to fairly reproduce the SIV anomaly. The results from Section 3.2 provide further elements to answer the other three open questions of this study, as follows:

**(ii) What are the best locations for in situ sampling of predictor variables to optimize the statistical predictability of SIV anomalies in terms of reproducibility and variability?**

The ten first best sampling locations were identified. The exact coordinates of these locations are provided in Table 3 and also plotted in Figs. 8 and 9. As suggested by the ensemble of model outputs, the 1st optimal location is placed at the transition Chukchi Sea–Central Arctic–Beaufort Sea (158.0°W, 79.5°N). The 2nd, 3rd and 4th best locations are placed near the North Pole (40.0°E, 88.5°N), at the transition Central Arctic–Laptev Sea (107.0°E, 81.5°N) and offshore the Canadian Archipelago (109.0°W, 82.5°N).

**(iii) How many optimal sites are needed for explaining a substantial amount (e.g., 70% – an arbitrarily chosen threshold) of the original SIV anomaly variance?**

By considering an arbitrary threshold of 70%, the systems AWI-LR (75%), AWI-HR (73%), HadGEM3-LL (79%) and HadGEM3-MM (74%) suggest that as few as four stations are sufficient to pass this threshold, that is also confirmed by the ensemble mean (73%). Even though the ECMWF predictors have slightly low skill, they are still not far from the threshold: ECMWF-LR (66%) and ECMWF-HR (64%). The ensemble mean indicates that five and six well-placed stations could explain about 75% and 80% of the SIV anomaly variance, respectively. Adding further to six well placed locations does not substantially improve the statistical predictability. Ten locations explain about 84% of the variance. However, as suggested by Fig. 8, even though the SEM seems to fairly reproduce the SIV anomaly variance and, therefore, the long-term variability, it found more difficulties to reproduce the short-term variabilities.

**(iv) Are the results model dependent, in particular, are they sensitive to horizontal resolution?**

The results suggest that statistical predictability is affected by model resolution. Notwithstanding, the question of whether or not a higher horizontal resolution provides better statistical predictability depends on the metric used to evaluate the predictions (Section 3.2.2 and Fig. 11). That is the case for RMSE, where the main target is to evaluate the reproducibility of the reconstructed values. It seems that an improved horizontal resolution allows a better trained statistical model so that the reconstructed values approach better to the original SIV anomaly (Fig. 11a). On the other hand, investigating the interannual variability, the predictors provided by the numerical models with lower resolution are more able to approach the reconstructed time series to the original SIV anomaly (Fig. 11b). As argued above, this study shows that model-based statistical predictability of SIV anomaly is sensible to the model horizontal resolution. Further investigation is needed to better understand the impact of model resolution on the SIV predictability.

## 5    Conclusions

We believe that this work positively impacts three aspects of a real-world observing system. First, by providing recommendations for optimal sampling locations. We are confident that our multi-model approach provides a solid view of the sites that better represent the variability of the pan-Arctic SIV. Second, even if those regions are not taken into account for any reason (for instance, logistic, environmental harshness, strategical sampling, etc), observationalists could still take advantage of the "region of influence" concept. By doing so, they avoid deploying two or more observational platforms that would provide relatively similar information in terms of pan-Arctic SIV variability. Third, considering that observational platforms are already operational, our SEM could be trained with model outputs (with the same or other state-of-the-art AOGCMs) and so fed with observational data to project future pan-Arctic SIV variability. Within this context, we expect that this manuscript will provide recommendations for the ongoing and upcoming initiatives towards an Arctic optimal observing design.

Despite these promising results, we recognize that it might be harder to achieve skillful predictions in the real-world employing statistical tools because the actual SIV variability is likely noisier than the one described by AOGCM outputs. While model results provide an average representation of variables inside a grid cell, real-world observations would be much more heterogeneous. This issue is even more pronounced when looking at our main predictor (SIT) due to the inherent roughness and short-scale spatial heterogeneity of the real-world SIT. As consequence, this heterogeneity may be a source of uncertainties in a real observing system and more observations would be required for effectively predict the SIV anomaly. Some caution should be exercised since our findings might be slightly different for other AOGCMs. A good perspective for addressing this issue is to reapply the methodology developed in this manuscript, but using all models that will be made available through the CMIP6. Also, with the sea ice depletion, some of the suggested optimal sampling locations might in the future be ice free.

Finally, it is worthwhile mentioning the recent effort from the scientific community to enhance the Arctic observational system. This effort takes place through recent observational programs such as the Year Of Polar Prediction (YOPP) (Jung et al., 2016) and the Multidisciplinary drifting Observatory for the Study of Arctic Climate (MOSAiC; https://www.mosaic-expedition.org/; last access: 01 March 2020).

*Author contributions.* LP, FM, and TF designed the science plan. DD computed the pan-Arctic sea ice area and volume. LP developed the statistical empirical model, conducted the data processing, produced the figures, analyzed the results, and wrote the manuscript based on the insights from all co-authors.

*Code availability.* All codes for computing and plotting the results of this article are written in Python programming language and are available upon request.

*Data availability.* All model outputs used in this study were made available through PRIMAVERA project (https://www.primavera-h2020.eu/; last access: 01 March 2020).

*Competing interests.* No competing interests are present.

*Acknowledgements.* The work presented in this paper has received funding from the European Union's Horizon 2020 Research and Innovation programme under grant agreements no. 727862 (APPLICATE project – Advanced prediction in Polar regions and beyond) and no. 641727 (PRIMAVERA project – PRocess-based climate sIMulation: AdVances in high-resolution modelling and European climate Risk Assessment). Leandro Ponsoni was funded by APPLICATE project until September 2019, and is now a Postdoctoral Researcher of the Fonds de la Recherche Scientifique (FNRS). François Massonnet is a FNRS Research Associate. David Docquier was funded by the EU Horizon 2020 PRIMAVERA project until September 2019, and is currently funded by the EU Horizon 2020 OSeaIce project, under the Marie Sklodowska-Curie grant agreement no. 834493. Guillian Van Achter is founded by PARAMOUR project which is supported by the Excellence Of Science programme (EOS), also founded by FNRS. We thank two anonymous reviewers and the editor, Dr. Petra Heil, for their constructive suggestions and criticism.

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
