# Peer review of "Statistical predictability of the Arctic sea ice volume anomaly: identifying predictors and optimal sampling locations"

_The Cryosphere, 2019_

## Referee Comment (RC1) · Anonymous Referee #1 · 9 Jan 2020

GENERAL OVERVIEW

The manuscript presents a statistical model for predicting the pan-Arctic Sea Ice Volume (SIV) anomaly on an interannual timescale. The long-term variability and the seasonal cycle have been subtracted to focus on the interannual SIV anomalies only, therefore excluding other better-understood signals. The statistical model is trained on the output of three coupled climate models produced in the frame of the HighResMIP. A low and high-resolution version of each model is analyzed.

The first part of the study inspects the capability of seven predictors to represent the sea ice volume up to 12 months in advance. The authors focus on two target months:

[Figure]

March (post-winter conditions) and September (late summer conditions). These predictors are tested and combined, both on a pan-Arctic and regional scale. The results show that the best predictive skill comes from the SIV itself, and by the Sea Ice Thickness (SIT), while the other considered variables are progressively less skillful.

The study presents afterward a method to determine some optimal locations that are representative of the SIV anomaly variance. Those locations are picked in a smart way to avoid clustering of points in certain regions, while other parts of the Arctic Ocean are underrepresented. The authors show that the statistical model can reconstruct approximately 70% of the SIV anomaly variance when fed with only 4 well-placed locations.

Even though the results here presented are in line with our expectations and not surprising, the manuscript tries to establish a robust protocol to predict the SIV anomaly. Furthermore, the fact that a large part of this variance can be predicted with only a few sparse observations in strategic locations is certainly interesting and can guide the design of future observation campaign in the Arctic region. The comparison of high and low resolutions contributes to the ongoing discussion in the modeling community about the benefit of resolving small features compared to the computational costs.

The approach followed by the authors as well as the application of this methodology to the SIV anomaly is quite novel. The purpose of the work is well presented and the methodology is adequately explained. The model data here analyzed are cutting edge in terms of model physics and resolution. The manuscript is well written and the figures and tables convey the message effectively.

The content of the study is certainly appropriate for The Cryosphere and I recommend the publication of this manuscript. Below I include a few minor points and suggestions that the authors should be able to address easily.

SPECIFIC COMMENTS

The manuscript provides several sampling locations with a multi-model approach. In

my understanding, these locations are computed based on annually-averaged fields. I am wondering if the sampling locations could be different for different target months. Also, some of the selected sampling locations might be ice-free in some periods of the year. Could the authors comment on this?

I believe that an interesting exercise would be comparing the performance of the statistical model in the optimal location to that in randomly chosen locations. This would show that the described method is robust and in fact, needed.

While the current model results provide an average representation of some variables inside a grid cell with a substantial extension, and the gradients between different cells are generally small, real-world observations would be much more localized and heterogeneous. Would this heterogeneity introduce some sampling errors and consequently require more observations to explain the SIV anomaly variance?

Is the whole time period ($\sim$150 years) necessary to reach the described results? I think it would be interesting to assess how many years of observations would be necessary to train adequately the statistical model here presented, and robustly reproduce the HighResMIP results.

1 – Line 16: It is worth mentioning also the SMOS sea ice thickness product.

2.1 – Line 6: Are the analysis on AWI-CM performed on the original FESOM2 grid or was the model output interpolated to a regular grid?

2.1 – Line 7: I would mention that the resolution difference between HR and LR in the Arctic is much lower in AWI-CM compared to the other two systems.

2.2 – Line 34: Is there a particular reason for choosing AWI-LR?

2.4 – Line 12: Be specific about the "common grid". Is it a low or high-resolution grid. Can this have an impact on the results?

---

## Referee Comment (RC2) · Anonymous Referee #2 · 14 Jan 2020

Statistical predictability of the Arctic sea ice volume anomaly: identifying predictors and optimal sampling locations. By L Ponsoni et al.

Summary statement The motivation for this study is to contribute to an Arctic observing system by identifying key locations where sea ice thickness should be measured in order to have predictability. This study contributes to predictability and also to a stakeholder need (i.e., observationalists) of developing an efficient Arctic observing network. I feel the science is strong with really interesting (and useful) results and worthy of publication. The figures are well-prepared and understandable. My main critique is that the text needs to be smoothed out and clarified. I made detailed suggestions through

about half of the document and these comments can be applied throughout the remaining parts of the paper. I have a few interpretation suggestions in the major comments. This paper is relevant for a broad science audience so the clarity of the writing is really critical for it to be broadly accessible.

Major Comments 1) It may be very useful to include stronger arguments as to why this is a model-only study. This can be strengthened in the introduction. (Page 3 lines 10-15, expand here in a way that puts it to rest). You want to be more convincing as to why this will be applicable in real life. 2) It needs to be made clear when the models are described (bottom page 3) that these are coupled climate models and are not pegged to observed conditions. Also, Discuss the GHG scenarios used for these particular simulations because all this information will make it easier for the reader to understand the results. For climate people, these are known but this paper should be accessible by weather and observational scientists as well as potentially policy experts (since they will help formulate the Arctic observing network). 3) Beginning of Section 2.2. This first paragraph lays out the methodology. I have read it twice and it is not easily understandable. Please revise this to be more precise and direct. I am not sure what to suggest specifically. Some thoughts a. Define anomaly earlier when you refer to fig 1. Just use it here. b. Move the sentence 'Overall , two categories of predictors are tested... (line 18, page 5) to be the second sentence. c. Revise the first sentence of your paragraph (your topic sentence) to something like: 'Potential predictor variables are identified for the empirical statistical model that predicts SIV anomalies.' There are extra words in this sentence and the key point of the paragraph is getting lost. 4) I have some suggestions regarding the structure of the writing. a. Strengthen your 'topic sentences' that start each paragraph. This sentence should tell the reader what is in this paragraph without having to read the paragraph. The sentences in the paragraph provide the evidence or facts to support the topic sentence. This type of structure makes it easier for the reader to understand your paper quickly. 5) It is not clear to me what the time scale for the predictions is in Section 2? (re: Fig 2, Table 1). It is one-month lead? Lag-0 is what I think it is but I did not see this explained clearly. In addition,

further interpretation of the panels in Fig. 2 would be helpful because reading the 2.2 and 2.3, which refer back to Fig. 2, I see that I do not have a clear understanding or appreciation for what Fig 2 shows. It would be good to discuss each panel and provide interpretation of the panel. 6) Could OHT be a poor predictor in these models because of model biases such as too strong stratification in the Arctic ocean so that 'heat' never makes it to the upper layers? This may be worthy of the discussion. 7) Conclusions. The results are summarized very nicely in the model context. As an observationalist (BTW, I am a modeler), I would want to know how this is relevant in the real world. Some discussion on linking this to observations would be nice. I know this is not easy and I do not suggest that you do this research for this paper, but providing these insights will help you link it better to the people you want to use this work. If you can provide a framework that links this study to the observations, that would really strengthen the paper.

Minor Comments 1) Page 1, Line 24, change 'proven to bring' to 'led to' 2) Page 2, Line 1, change 'disturbance of' to 'disturbance in' 3) Page 2, line 1, split everything 'which has also . . .' into a separate sentence to make it easier to understand. 4) Page 2, line 4, change 'sailing routes' to 'ship routes', not all of the ship may be sailboats. 5) Page 2, line 5, change 'At global scale' to 'Globally' 6) Page 3, line 1, change 'To the knowledge of the authors' to 'To the best of the authors' knowledge' 7) Page 3, line 15-16, change 'What are the performance ..' to 'What is the performance. . .' 8) Page 3, line 17, change 'a large amount . . .' to 'a substantial (e.g., 70%) of the original..' 9) Page 4, Figure 1 top panel is not even mentioned in the text. The figure panels have a and b on the right-hand side. I did not see them at first. It is standard to have them on the left corner. I suggest you edit this on all your figure panels. 10) Page 4, line 3, make it clear that the long-term trend and seasonal cycle has been removed. The text '(no long-term trend; no seasonal cycle)' is somewhat vague. Was there never a trend? It is clear from the top panel that there are trends but it is helpful for the reader if the language is unambiguous. 11) Page 5, line 7, Clarify the geographical span of the different resolution. For a student, the changing resolution is confusing. 12) Section
2.3 is written very clearly. It may be worth saying something about including SIV in the SEM. IT seems to me that SIV could dominate the results since the autocorrelation is so strong in SIV. 13) Section 2.4, numerous grammar issues in this section. This section is rough and needs revision. 14) Fig. 10, the lag/lead time for the reconstruction is not clear to me, related to comments about Fig. 1. 15) Page 22, line 30, remove 'respectively'. I do not think that is needed here because the numbers are the same as highlighted by the word 'both'.

---

## Author Response (AR1)

Dear Editor,

Thank you for the time that you have spent on our manuscript. We are happy with your feedback and grateful for your comments and suggestions. In addition to the previously answered referee's comments, below you will find a summary of the changes that we have made throughout the manuscript to address all of your suggestions. The replies to your comments are written in blue, while your comments are reproduced in **black**.

Attached to this letter, you will find the marked-up version of the manuscript. The marked-up version highlights the changes that we have done regarding the first manuscript's version. Please, notice also that due to your comments below, the line, page, and figure numbers mentioned in our rebuttal letters to the referees have now slightly changed. To avoid confusion, we are also attaching an updated version of the rebuttal letters in which we are indicating the page and lines for each referees' comment in **green**.

Yours sincerely and on behalf of all co-authors,

Leandro Ponsoni
* * *
**Editor Decision: Publish subject to minor revisions (review by editor) (15 Apr 2020) by Petra Heil**

**Comments to the Author:**

Dear Dr Ponsoni.

Thank you for ypur submission tc-2019-257 to TC.

"Statistical predictability of the Arctic sea ice volume anomaly: identifying predictors and optimal sampling locations"

Pls review the comments of both reviewers carefully and address these together with the ones outlined here in your revised version.

Editor's general comments:

\* End of Abstract: Add info on how much of the SIV var anomaly is achieved by six well-placed data locations.
In the new manuscript's version, the abstract is slightly reformulated and it contains this information.

\* 1-24: Add what is impacted: "to bring significant impacts" --> To what? Then connect to the follow-on effects (your "Regionally, native"). --> You might need to bring your "global" impacts forward and then discuss the "regional" ones, as the former are related to the climate system, the others are follow
We have improved the first paragraph of the manuscript. Among other improvements, we are now bringing more "impacts" to the discussion. As you suggested, we have first addressed the global impacts and so the regional ones. That was indeed a good point. Thank you for pointing it out.

\* 2-10: "meltdown" is very strong and inappropriate wording. Suggest to change, i.e., "intense sea-ice loss" or "rapid sea-ice loss".

We have replaced "meltdown" by "intense sea ice loss".

\* 2-19: Swart et al., [2015] are really about the relevance of internal variability in coupled climate (CMIP style) models. The statement "it has been already shown that trends in the pan-Arctic sea ice extent can be masked by its long-term variability (Swart et al., 2015)" is not correct. Pls rephrase.

We have revisited Swart et al. [2015] and we agree with this comment. We have reformulated the paragraph. This sentence was dropped from the text since it does not bring impact to the paragraph's content.

\* There are some resolved issues on the underperformance of OHT, especially in conjunction with SST as predictant. Would you kindly expand in the "Discussion" section?

In the new version of the manuscript, there is a full paragraph dedicated to this point in Sec. 4 (Discussion).

\* The early part of your paper promises an analysis on how spatial model resolution affects the SIV prdictablity, but the manuscript do not follow through on this. Pls explore this and discuss in the "Discussion" section.

In the introduction of the manuscript, we have raised the question of whether or not the results are model dependent, in particular, whether they are sensitive to horizontal resolution. We have shown that indeed model resolution has an impact on the results. Depending on the analyzed metrics, the resolution impacts positively or negatively the statistical prediction. However, at this stage, we do not have a clear understanding of this point, and further investigation is needed to better understand the impact of model resolution on the SIV statistical predictability. We have highlighted all these aspects in the discussions section (Sec. 4).

\* Pls separate the "Discussion" section from "Conclusion" section.

Done.

Minor comments:

1-2: Change "6" and "3" to "six" and "three".

Done.

1-3: Change "2" to "two".

Done.

1-6: Italize "in situ". -- Throught manuscript.

Done for all instances in the text.

1-10: Change "4" to "four".

Done.

1-11: Change "enough" to "sufficient".

Done.

1-14: Change "6" to "six".

Done.

1-14: Change "As per 6 well-placed locations" to read "Adding further to six well place locations" and remove "by adding new sites".
Done.

1-15: Change "4" to "four"... Pls change all numericals (in text), that are less than twelve to "words".
Done. Also for the entire manuscript.

2-29: Remove "trivial".
Done.

2-31: Replace "for feeding" with "as input into".
Done, also in other instance throughout the text.

3-11: Replace "broadly" with "well"
Done.

3-20ff: Remove "Following this introduction ... observing sampling design."
Done.

3-30: Assuming that the horizontal resolution (and subsequently bathymetry, inputs, etc associated with the grid res) is the only difference, then:
a) Replace "model configurations" with "model horizontal grids", and
b) Remove "These configurations differ by their horizontal grid resolution (in both the atmosphere and ocean)." (line 3-31).
Done. Please, notice that this section was slightly reformulated to address the referees' comments.

4-3: Figure references in the body of a manuscript should not be a repeat of the figure caption. Instead figures should be referenced in the text to support a statement/description of the displayed parameter(s). --> Rephrase.
We agree with the editor. This issue is corrected in the new version of the manuscript.

4-24: Remove "best".
Done (I guess pg. 5, though).

4-25: Replaece "non–significant" with "insignificant".
Done (I guess pg. 5, though).

6-1: Regarding reference to "Table 1", pls refer to my comment at 4-3.
Solved issue.

6-Tab1: In caption remind the reader why there is no CorrCoef for OHT and SIV for the high-res models.
Done.

7-Fig2: The order of the sub-figures is illogical, pls relabel.
Done.

8-19: Change "at least not" to "especially not".
Please, notice that we have followed the recommendation of Referee #2 and we reformulated Sec. 2.4. Due to that, this change is no longer required.

8-30: Change "to feed the Eq. 4" to "in Eq. 4".
Done.

9-15: Change "we borrow the concept" to "we follow the concept".
Done.

9-25: Change "Figure 3 shows how would be the region of influence ... ensemble of datasets." to "The region of influence for a station arbitrarily placed at the North Pole, as defined by the ensemble of datasets, exhibits depatures from concentric reflecting the transpolar drift. (Fig. 3)."
This suggestion was incorporated into the text, but please notice that we have reformulated Sec. 2.4 as suggested by Referee #2.

14-4: Change "6 models" to "sixe model realisations".
Done.

16-Fig7: Why does the colour scheme for each subfig in the right hand column change? Please show for a single colour scheme.
The color scheme used in the right-hand column in Fig. 7 is coherent with the representation of the same regions of influence in Fig. 8. If the editor allows, we would like to keep in this way so that the comparison between Fig. 7 and Fig. 8 is straightforward.

22-29: Change "ALL" to "all".
Done. Also in other instances throughout the manuscript.

23-2: Change "by observing platforms" to read "by autonomous observing platforms".
Done.

23-4: Change "The first" to "The former".
Done.

23-4: Change "not turn out to" to "did not act as".
Done.

23-4: Change "predictor (at least 5 not when using monthly means)." to "predictor, at least not when using monthly means."
Done.

23-19: Change "that only 4 stations are enough to overpass" to "that as few as four stations are sufficient to pass".
Done.

24-7: Change "might be in a free-ice region in the future." to "might in the future be ice free."
Done.

24-11: Spell out "MOSAiC".
Done.

**Statistical predictability of the Arctic sea ice volume anomaly: identifying predictors and optimal sampling locations**

Leandro Ponsoni[1,2], François Massonnet[1,2], David Docquier[3], Guillian Van Achter[1], and Thierry Fichefet[1]

[1]Georges Lemaître Centre for Earth and Climate Research (TECLIM), Earth and Life Institute, Université catholique de Louvain, Louvain-la-Neuve, Belgium
[2]Fonds de la Recherche Scientifique – FNRS, Belgium
[3]Rossby Centre, Swedish Metereological and Hydrological Institute, Norrköping, Sweden

*Correspondence to:* Leandro Ponsoni (leandro.ponsoni@uclouvain.be)

**Abstract.** This work evaluates the statistical predictability of the Arctic sea ice volume (SIV) anomaly – here defined as the detrended and deseasonalized SIV – on the interannual time scale. To do so, we made use of  six datasets, from  three different atmosphere-ocean general circulation models, with  two different horizontal grid resolutions each. Based on these datasets, we have developed a statistical empirical model which in turn was used to test the performance of different predictor variables, as well as to identify optimal locations from where the SIV anomaly could be better reconstructed and/or predicted. We tested the hypothesis that an ideal sampling strategy characterized by only a few optimal sampling locations can provide  *in situ* data for statistically reproducing and/or predicting the SIV interannual variability. The results showed that, apart from the SIV itself, the sea ice thickness is the best predictor variable, although total sea ice area, sea ice concentration, sea surface temperature, and sea ice drift can also contribute to improving the prediction skill. The prediction skill can be enhanced further by combining several predictors into the statistical model. Feeding the statistical model with predictor data from  four well-placed locations is  sufficient for reconstructing about 70% of the SIV anomaly variance. ~~An improved model horizontal resolution allows a better trained statistical model so that the reconstructed values approach better to the original SIV anomaly. On the other hand, if we look at the interannual variability, the predictors provided by numerical models with lower horizontal resolution perform better when reconstructing the original SIV variability. As per 6 well-placed locations, the statistical predictability does not substantially improve by adding new sites.4~~ four first best locations are placed at the transition Chukchi Sea–Central Arctic–Beaufort Sea (158.0°W, 79.5°N), near the North Pole (40°E, 88.5°N), at the transition Central Arctic–Laptev Sea (107°E, 81.5°N), and offshore the Canadian Archipelago (109.0°W, 82.5°N), in this respective order. Adding further to six well placed locations, which explains about 80% of the SIV anomaly variance, the statistical predictability does not substantially improve taking into account that ten locations explain about 84% of that variance. An improved model horizontal resolution allows a better trained statistical model so that the reconstructed values approach better to the original SIV anomaly. On the other hand, if we look at the interannual variability, the predictors provided by numerical models with lower horizontal resolution perform better when reconstructing the original SIV variability. We believe that this study provides recommendations for the ongoing and upcoming observational initiatives, in terms of an Arctic optimal observing design, for studying and predicting not only the SIV values but also its interannual variability.

**1 Introduction**

The continuous melting of the Arctic sea ice observed in the last decades (e.g., **?????????**), associated with the respective reduction in total sea ice area (SIA) and volume (SIV), has led to significant impacts at global and regional scales. Globally, the sea ice depletion is reported to impact some aspects of the weather at low- and mid-latitude regions, by means of both oceanographic (**??**) and atmospheric teleconnections (**??**), including the higher occurrence of extreme events (**????**). Regionally, high-trophic predators such as seabirds (**??**) and mammals (**?????**) are changing 
[revised manuscript text omitted]
 envisage three main ways by which this work could support observationalists in a real-world observing system. The first is providing recommendations for optimal sampling locations. We believe that our multi-model approach provides a solid view of the sites that better represent the variability of the pan-Arctic SIV. Second, even if those regions are not taken into account for any reason (for instance, logistic, environmental harshness, strategical sampling, etc), observationalists could still take advantage of the "region of influence" concept. By doing so, they avoid deploying two or more observational platforms

25 that would provide relatively similar information in terms of pan-Arctic SIV variability. Third, considering that observational platforms are already operational, our SEM could be trained with model outputs (with the same or other state-of-the-art AOGCMs) and so fed with observational data to project future pan-Arctic SIV variability. Within this context, we expect that this manuscript will provide recommendations for the ongoing and upcoming initiatives towards an Arctic optimal observing design.

30 Despite these promising results, we recognize that it might be harder to achieve skillful predictions in the  real-world employing statistical tools because the actual SIV variability is likely noisier than the one described by AOGCM outputs. While model results provide an average representation of variables inside a grid cell, real-world observations would be

much more heterogeneous. This issue is even more pronounced when looking at our main predictor (SIT) due to the inherent roughness and short-scale spatial heterogeneity of the real-world SIT. As consequence, this heterogeneity may be a source of uncertainties in a real observing system and more observations would be required for effectively predict the SIV anomaly. Some caution should be taken since our findings could be slightly different for other AOGCMs. A good perspective for addressing this issue is to reapply the methodology developed in this manuscript, but using all models that will be made available through the CMIP6. Also, with the sea ice depletion, some of the optimal sampling locations here suggested might  in the future be ice free.

Finally, it is worthwhile mentioning the recent effort from the scientific community to enhance the Arctic observational system. This effort takes place through recent observational programs such as the Year Of Polar Prediction (YOPP) (**?**) and the Multidisciplinary drifting Observatory for the Study of Arctic Climate (MOSAiC; https://www.mosaic-expedition.org/; last access:  01 March 2020).

*Author contributions.* LP, FM, and TF designed the science plan. DD computed the pan-Arctic sea ice area and volume. LP developed the statistical empirical model, conducted the data processing, produced the figures, analyzed the results, and wrote the manuscript based on the insights from all co-authors.

*Data availability.* All model outputs used in this study were made available through PRIMAVERA project (https://www.primavera-h2020.eu/; last access: 01 March 2020).

*Competing interests.* No competing interests are present.

*Acknowledgements.* The work presented in this paper has received funding from the European Union's Horizon 2020 Research and Innovation programme under grant agreement no. 727862: APPLICATE project (Advanced prediction in Polar regions and beyond). Leandro Ponsoni was funded by APPLICATE project until September 2019, and is now a Postdoctoral Researcher of the Fonds de la Recherche Scientifique – FNRS. François Massonnet is a Research Associate of FNRS. David Docquier  was funded by the EU Horizon 2020 PRIMAVERA project  (grant agreement no. 641727) until September 2019, and is currently funded by the EU Horizon 2020 OSeaIce project, under the Marie Sklodowska-Curie grant agreement no.  834493. Guillian Van Achter is founded by PARAMOUR project which is supported by the Excellence Of Science programme (EOS), also founded by FNRS.  We thank two anonymous reviewers and the editor, Dr. Petra Heil, for their constructive suggestions and criticism.

Dear Referee,

Thank you for the time that you have spent on our manuscript. We are happy with your positive response and grateful for your comments and suggestions. These certainly contributed to improving the quality of our manuscript.

Below you will find a summary of the changes that we have made throughout the manuscript to address all of your suggestions. The replies to your comments are written in blue, while your comments are reproduced in black. Please, notice that line, page, and figure numbers mentioned in our rebuttal letter refer to the new version of the manuscript unless stated otherwise.

Yours sincerely and on behalf of all co-authors,

Leandro Ponsoni
* * *
**Anonymous Referee #1**

GENERAL OVERVIEW

The manuscript presents a statistical model for predicting the pan-Arctic Sea Ice Volume (SIV) anomaly on an interannual timescale. The long-term variability and the seasonal cycle have been subtracted to focus on the interannual SIV anomalies only, therefore excluding other better-understood signals. The statistical model is trained on the output of three coupled climate models produced in the frame of the HighResMIP. A low and high-resolution version of each model is analyzed.

The first part of the study inspects the capability of seven predictors to represent the sea ice volume up to 12 months in advance. The authors focus on two target months: March (post-winter conditions) and September (late summer conditions). These predictors are tested and combined, both on a pan-Arctic and regional scale. The results show that the best predictive skill comes from the SIV itself, and by the Sea Ice Thickness (SIT), while the other considered variables are progressively less skillful.

The study presents afterward a method to determine some optimal locations that are representative of the SIV anomaly variance. Those locations are picked in a smart way to avoid clustering of points in certain regions, while other parts of the Arctic Ocean are underrepresented. The authors show that the statistical model can reconstruct approximately 70% of the SIV anomaly variance when fed with only 4 well-placed locations.

Even though the results here presented are in line with our expectations and not surprising, the manuscript tries to establish a robust protocol to predict the SIV anomaly. Furthermore, the fact that a large part of this variance can be predicted with only a few sparse observations in strategic locations is certainly interesting and can guide the design of future observation campaign in the Arctic region. The comparison of high and low resolutions contributes to the ongoing discussion in the modeling community about the benefit of resolving small features compared to the computational costs.

The approach followed by the authors as well as the application of this methodology to the SIV anomaly is quite novel. The purpose of the work is well presented and the methodology is adequately

explained. The model data here analyzed are cutting edge in terms of model physics and resolution. The manuscript is well written and the figures and tables convey the message effectively.

The content of the study is certainly appropriate for The Cryosphere and I recommend the publication of this manuscript. Below I include a few minor points and suggestions that the authors should be able to address easily.

Again, we thank the referee for her/his time and detailed revision of our manuscript. We appreciated very much her/his comments, which were all taken into account in the revised version of the paper. Below, we answer point-by-point all specific comments.

SPECIFIC COMMENTS

The manuscript provides several sampling locations with a multi-model approach. In my understanding, these locations are computed based on annually-averaged fields. I am wondering if the sampling locations could be different for different target months. Also, some of the selected sampling locations might be ice-free in some periods of the year. Could the authors comment on this?
All results of the manuscripts are based on monthly-averaged fields. This point is clarified in the text [pg. 4, l.18–19] **[pg. 4, l. 29–30]**.

Except from Sec. 3.1, where we first assessed the performance of different predictors by focusing on March and September (Figs. 4 and 5), the other sections do not make a distinction of months.

However, we understand the referee's comments since we had posed the same question to ourselves during the preparation of the manuscript's first version. We have decided to avoid the distinction of months for the following reasons:

i.   The motivation of the manuscript is to provide support for a **year-round *in situ* monitoring system**. Thus, those are sampling locations that better reproduce/predict the pan-Arctic sea ice volume taking into account continuous monitoring throughout the entire year.

ii.  A distinction of months would likely suggest relatively different locations. In the real world, **this would require a re-positioning of observational platforms** (e.g., moorings and/or buoys) every month.

iii. The fact that some sampling locations might be ice-free in some periods of the year **is part of the time-series variability and it brings predictability to the statistical model as well**. If the grid-point is ice-free for long periods, predictors as SIT, SIC and Drift will be disregarded by the correlation map criterion. The SST predictor can still be useful even from grid-cells which are mostly of the year ice-free. Nevertheless, the four most performance locations are likely covered by sea ice during the entire year, for most of the years.

iv.  By splitting the time series into 12 parts, we substantially **reduce the number of points for training and applying the statistical model**. The fact that the statistical model is randomly trained (70% of the data) and applied (30% of the data) within a Monte Carlo (MC) scheme (500 reproductions) give us statistical robustness to assume that this configuration is the best scenario for a year-round sampling system. We have tried to increase the number of MC interactions but it turned out that 500 is already a safe threshold.

I believe that an interesting exercise would be comparing the performance of the statistical model in the optimal location to that in randomly chosen locations. This would show that the described method is robust and in fact, needed.

We absolutely agree, thanks for the interesting suggestion. To address it, we have compared the RMSE and R2 calculated between the original and our-methodology-based reconstructed SIV anomalies (as shown in Fig. 11a,b) against the same two metrics estimated by randomly chosen locations. To do so, we have determined 100 combinations of 10 randomly chosen locations. For each combination, we reconstructed the SIV anomaly using data from the 1st location, the 1st–2nd, the 1st–3rd, the 1st–4th, and so on. Fig. A (this rebuttal letter) shows 2 of the 100 sets of randomly chosen locations. For the sake of fairness, we have used only predictors from grid points enclosed into the region highlighted by the red line in Fig. A. This region represents our global region of influence as defined by Fig. 8 (now this line is also plotted in Fig. 9). It is worthwhile saying that 100 combinations of randomly locations already provide robust statistics for such a comparison.

Fig. B shows that the SIV reconstructions based on our methodology (and optimally selected locations) are more skillful compared to the predictions provided by the randomly chosen locations, taking into account both metrics (RMSE and R2). This is valid for all models, considering a single location and/or any combination of 1 up to 10 locations.

These results, and respective supporting Fig. B, were incorporated into the new version of the manuscript  **[pg. 23, l. 1–9, Fig. 12]**.

While the current model results provide an average representation of some variables inside a grid cell with a substantial extension, and the gradients between different cells are generally small, real-world observations would be much more localized and heterogeneous. Would this heterogeneity introduce some sampling errors and consequently require more observations to explain the SIV anomaly variance?

As the referee highlighted, real-world observations are much more heterogeneous than averaged grid cell values. Compared to other oceanographic parameters such as temperature and salinity (unless in regions marked by steep frontal systems), this issue is even more pronounced when looking at the sea ice due to its inherent roughness. Thus, we indeed expect that this heterogeneity may be a source of uncertainties in a real observing system. We also agree that more observations could attenuate these uncertainties. This is a very important point that was quickly addressed in the first manuscript's version [former pg. 24, lines 3–4]. We have added a few more words to make this point clear in the manuscript  **[pg. 26, l. 26–31]**.

Is the whole time period (~150 years) necessary to reach the described results? I think it would be interesting to assess how many years of observations would be necessary to train adequately the statistical model here presented, and robustly reproduce the HighResMIP results.

We have used model outputs from coupled historical runs, referred to as "hist-1950", performed within the context of HighResMIP. So, from all model configurations the data spans about 65 years, starting in the early 1950s and finishing in mid-2010s  **[pg. 3, l. 33–34]**. We understand that using these 65-years is indeed necessary to achieve statistical robustness.

1 – Line 16: It is worth mentioning also the SMOS sea ice thickness product.

SMOS is now mentioned in the text  **[pg. 3, l. 15]**.

2.1 – Line 6: Are the analysis on AWI-CM performed on the original FESOM2 grid or was the model output interpolated to a regular grid?

**Sea-ice concentration (SIC)** was provided by the AWI group on the original atmosphere grid in the framework of the PRIMAVERA project. **Sea-ice thickness (SIT)**, **sea-surface temperature (SST)** and **sea-ice drift speed (Drift)** were also provided by AWI but on a 1-degree regular grid also in the framework of the PRIMAVERA project. **Sea-ice area (SIA)** was computed from the SIC files and the atmosphere grid-cell area, while **Sea-ice volume (SIV)** was computed from the SIT files and the ocean grid-cell area (Docquier et al., 2019). Finally, ocean heat transport (OHT) was computed by the AWI group directly from the raw data. Additional information is presented in Section 2.1 of Docquier et al. (2019).

2.1 – Line 7: I would mention that the resolution difference between HR and LR in the Arctic is much lower in AWI-CM compared to the other two systems.

This is indeed a good point. We thank the reviewer for spotting that. While the ocean resolution in AWI-LR and AWI-HR varies between 24 and 110km, and between 10 and 60km, respectively (with higher resolution in dynamically active regions), the ocean resolution is almost similar in the Arctic Ocean (~25km). We brought this information to the text [pg. 4, l. 33–34 to pg. 5, l.1–4] **[pg. 4, l. 21–26]**. In addition, since the grid used by AWI is not trivial to understand without a supporting plot, we are directing the reader to Fig. 4 of Sein et al. (2016).

2.2 – Line 34: Is there a particular reason for choosing AWI-LR?

No, there isn't a particular reason for choosing AWI-LR. We selected AWI-LR as the example-case. "AWI" is the first model in our alphabetically-sorted list and, in the other model-comparative figures (e.g., Fig. 6), we always referred first to the low resolution (LR) version. We clarified this point in Fig. 2's caption.

2.4 – Line 12: Be specific about the "common grid". Is it a low or high-resolution grid. Can this have an impact on the results?

We agree that this point requires clarification. As suggested by the score maps in Fig. 6, each model configuration indicates its own best sampling location (smallest RMSE in the score map). However, the RMSE values show that overall there is a good agreement on the regions with high scores (small RMSE values represented by yellow shades). To achieve an ensemble best location we first applied Eq. 4 to normalize all score maps between 0 and 1 so that the models have the same weight in the averaging step (Fig. C, first column). However, since the models have different grid-resolution, we have interpolated the score maps from the different models into a common 1°×1° grid. By performing this step, we can calculate an ensemble mean score map.

**The interpolation of the individual score maps into a common 1°×1° grid for further computation of an ensemble mean score map has no impact on the results** [pg. 9, l. 32–33] **[pg. 9, l. 28–29].**

Notice that the interpolated score maps (Fig. C, second column) preserve the best performance regions.

[Figure]

**Fig. A:** Map displaying two examples (out of 100) of randomly chosen locations. All random locations are placed into the area enclosed by the red line. This region represents our global region of influence as defined in Fig. 8.

[Figure]

**Fig. B:** Root Mean Squared Error (RMSE; left column) and coefficient of determination (R²; right column) calculated between the original and reconstructed SIV anomalies. The reconstructed SIV volume anomalies are based on the optimally selected locations following our methodology (full dots), as well as by randomly chosen locations (empty dots). In the last case, 100 sets of 10 randomly chosen locations are used. For each of the 100 sets, the SIV anomaly is reconstructed using data from the 1st location, the 1st–2nd, the 1st–3rd, ..., the 1st–10th. The random locations are all placed into the region enclosed by the red line shown in Fig. A. The vertical bars associated with the empty dots represent the one standard deviation.

[Figure]

**Fig. C:** Normalized score maps calculated for the different model outputs with the original grid (left column) and after the interpolation to the common 1°×1° grid (right column). Notice that the interpolation has no impact on the best performance regions (shades of yellow). The interpolation is a required step for calculating an ensemble score map since the models have different resolutions.

Dear Referee,

Thank you for the time that you have spent on our manuscript and for the detailed "Referee comment" report. We are happy with the positive response and grateful for your comments and suggestions. These certainly contributed to improving the quality of our manuscript.

Below you will find a summary of the changes that we have made throughout the manuscript to address all of your suggestions. The replies to your comments are written in blue, while your comments are reproduced in black. Please, notice that all line, page and figure numbers mentioned in our rebuttal letter refer to the new version of the manuscript, unless stated otherwise.

Yours sincerely and on behalf of all co-authors,

Leandro Ponsoni
* * *
**Anonymous Referee #2**

Summary statement The motivation for this study is to contribute to an Arctic observing system by identifying key locations where sea ice thickness should be measured in order to have predictability. This study contributes to predictability and also to a stake-holder need (i.e., observationalists) of developing an efficient Arctic observing network. I feel the science is strong with really interesting (and useful) results and worthy of publication. The figures are well-prepared and understandable. My main critique is that the text needs to be smoothed out and clarified. I made detailed suggestions through about half of the document and these comments can be applied throughout the remaining parts of the paper. I have a few interpretation suggestions in the major comments. This paper is relevant for a broad science audience so the clarity of the writing is really critical for it to be broadly accessible.

Again, we thank the referee for her/his thorough review of the manuscript. We appreciated very much her/his detailed comments not only in terms of science but also regarding the writing style. Below, we answer point-by-point all major and minor comments.

Major Comments

1) It may be very useful to include stronger arguments as to why this is a model-only study. This can be strengthened in the introduction. (Page 3 lines 10-15, expand here in a way that puts it to rest). You want to be more convincing as to why this will be applicable in real life.
We have slightly changed the introduction to properly address this point. Now we have reinforced in this part of the text that observations of sea ice thickness, required for calculating the SIV, present limitations in the warmer seasons. Therefore, this variable is not made available year-round from the classical satellite campaigns [pg. 3, l. 17–19] **[pg. 3, l. 16–18]**. In the following paragraph [pg. 3, l. 20–23] **[pg. 3, l. 18–20]**, we reemphasize that the models used in this work, which are cutting edge in terms of model physics and resolution, fairly represent the thermodynamic and dynamic sea ice processes linking predictors and predictand. In Sec. 4 5, we have added a discussion on how our study could be used in different ways by observationalists [pg. 26, l. 26–33] **[pg. 26, l. 17–25]**.

2) It needs to be made clear when the models are described (bottom page 3) that these are coupled climate models and are not pegged to observed conditions. Also, Discuss the GHG scenarios used for these particular simulations because all this information will make it easier for the reader to understand the results. For climate people, these are known but this paper should be accessible by weather and observational scientists as well as potentially policy experts (since they will help formulate the Arctic observing network).

That is indeed a good point. These two aspects are now clarified in the first paragraph of Sec. 2.1. [pg. 4, l. 3–13] **[pg. 3, l. 29 to pg. 4, l. 6]**.

3) Beginning of Section 2.2. This first paragraph lays out the methodology. I have read it twice and it is not easily understandable. Please revise this to be more precise and direct. I am not sure what to suggest specifically. Some thoughts a. Define anomaly earlier when you refer to fig 1. Just use it here. b. Move the sentence 'Overall , two categories of predictors are tested...(line 18, page 5) to be the second sentence. c. Revise the first sentence of your paragraph (your topic sentence) to something like: 'Potential predictor variables are identified for the empirical statistical model that predicts SIV anomalies.' There are extra words in this sentence and the key point of the paragraph is getting lost.

We agree with the referee. All paragraphs from Sec. 2.2 were rewritten to bring clarity to the text. To make it easier for the reader, an explanation for the term "anomaly" is provided in the Introduction [pg. 2, l. 31–32] **[pg. 2, l. 30]** and also in Sec. 2.1 [pg. 4, 20–22] **[pg. 5, 1–3]**.

4) I have some suggestions regarding the structure of the writing. a. Strengthen your 'topic sentences' that start each paragraph. This sentence should tell the reader what is in this paragraph without having to read the paragraph. The sentences in the paragraph provide the evidence or facts to support the topic sentence. This type of structure makes it easier for the reader to understand your paper quickly.

We thank the referee for the suggestion. We minutely addressed all the comments in this report taking into account this comment (4) and also the summary statement. We have promoted several changes throughout the text in order to make it clearer and easier to read for a non-specialized audience. Regarding this, we have asked for a few colleagues from different science fields to check whether or not the manuscript is understandable. Apparently, we have made the job. In any case, further comments on how to make this paper more accessible for a broader audience are always welcome.

5) It is not clear to me what the time scale for the predictions is in Section 2? (re: Fig 2, Table 1). It is one-month lead? Lag-0 is what I think it is but I did not see this explained clearly. In addition, further interpretation of the panels in Fig. 2 would be helpful because reading the 2.2 and 2.3, which refer back to Fig. 2, I see that I do not have a clear understanding or appreciation for what Fig 2 shows. It would be good to discuss each panel and provide interpretation of the panel.

It is indeed a lag-0 correlation. This is now clarified in the text [pg. 6, l. 13; Fig. 2's caption] **[pg. 6, l. 8; Fig. 2's caption]**. As mentioned in the answer to item (3), we have rewritten Sec. 2.2. In the new text, we are providing a better explanation of Fig. 2, considering all panels.

6) Could OHT be a poor predictor in these models because of model biases such as too strong stratification in the Arctic ocean so that 'heat' never makes it to the upper layers? This may be worthy of the discussion.

We agree with the referee. This might be a potential reason why OHT is a poor predictor. This is an interesting point that could be investigated further with more detailed analysis. We brought this discussion to the text [pg. 23, l. 25–28] **[pg. 23, l. 28–30]**.

7) Conclusions. The results are summarized very nicely in the model context. As an observationalist (BTW, I am a modeler), I would want to know how this is relevant in the real world. Some discussion on linking this to observations would be nice. I know this is not easy and I do not suggest that you do this research for this paper, but providing these insights will help you link it better to the people you want to use this work. If you can provide a framework that links this study to the observations, that would really strengthen the paper.

We envisage three main ways by which this work could support observationalists in a real-world observing system. The first is providing recommendations for optimal sampling locations. We believe that our multi-model approach provides a solid view of the sites that better represent the variability of the pan-Arctic SIV. Second, even if those regions are not taken into account for any reason (e.g., logistic, environmental harshness, etc), observationalists could still take advantage of the "region of influence" concept. By doing so, they avoid deploying two or more observational platforms that would provide relatively similar information in terms of pan-Arctic SIV variability. Third, considering that observational platforms are already operational, our SEM could be trained with model outputs (with the same or other state-of-the-art AOGCMs) and so fed with observational data to project future pan-Arctic SIV variability.

This discussion is now added to Sec. 45 [pg. 26, l. 26–33] **[pg. 26, l. 17–25]**.

Minor Comments
1) Page 1, Line 24, change 'proven to bring' to 'led to'
Changed [pg. 2, l. 5].

2) Page 2, Line 1, change 'disturbance of' to 'disturbance in'
Changed [pg. 2, l. 8] **[pg. 2, l. 11]**.

3) Page 2, line 1, split everything 'which has also...' into a separate sentence to make it easier to understand.
We have slightly reformulated the paragraph to accommodate this suggestion [pg. 2, l. 5–8] **[pg. 2, l. 8–12]**.

4) Page 2, line 4, change 'sailing routes' to 'ship routes', not all of the ship may be sailboats.
Changed [pg. 2, l. 9] **[pg. 2, l. 13]**.

5) Page 2, line 5, change 'At global scale' to 'Globally'
Changed [pg. 2, l. 11] **[pg. 5, l. 5]**.

6) Page 3, line 1, change 'To the knowledge of the authors' to 'To the best of the authors' knowledge'
Changed [pg. 3, l. 4] **[pg. 3, l. 3]**.

7) Page 3, line 15-16, change 'What are the performance ..' to 'What is the performance...'
Changed [pg. 3, l. 23] **[pg. 3, l. 22]**.

8) Page 3, line 17, change 'a large amount...' to 'a substantial (e.g., 70%) of the original..'
We have incorporated this suggestion, but in a slightly different way. We keep the info that 70% is an arbitrarily chosen threshold [pg. 3, l. 25] **[pg. 3, l. 24]**.

9) Page 4, Figure 1 top panel is not even mentioned in the text. The figure panels have a and b on the right-hand side. I did not see them at first. It is standard to have them on the left corner. I suggest you edit this on all your figure panels.

We have made a proper reference to Fig. 1a. in the text  **[pg. 4, l. 31–35]**. The panel index letter is now placed in the right-hand side in Fig. 1. Also in Figs. 4, 5, 11 and 12.
For Figs. 2, 6, 7 and 8 we preferred to keep the letter indicating the panel index centralized. We think the letters are easily spotted in that way.

10) Page 4, line 3, make it clear that the long-term trend and seasonal cycle has been removed. The text '(no long-term trend; no seasonal cycle)' is somewhat vague. Was there never a trend? It is clear from the top panel that there are trends but it is helpful for the reader if the language is unambiguous.
We agree with the referee. Indeed the text *"(no long-term trend; no seasonal cycle)"* is somewhat vague and confusing. We excluded this piece of text (or similar) from the entire manuscript. In the new manuscript's version, we have first defined SIV anomaly in the Introduction  **[pg. 2, l. 30]**. This definition is recalled when describing Fig. 1 in Sec. 2.1  **[pg. 5, 1–3]**.

11) Page 5, line 7, Clarify the geographical span of the different resolution. For a student, the changing resolution is confusing.
Indeed, this is an important point. The ocean resolution of **AWI-LR** varies from **24 to 110 km**, with **∼25 km in the Arctic**. The ocean resolution of **AWI-HR** varies from **10 to 60 km**, with a refined resolution in dynamically active regions (e.g., ∼10km in the vicinity of the Gulf Stream), and **∼25 km in the Arctic**.
An important point recalled by Referee #1 is that the resolution difference between HR and LR in the Arctic is much lower in the AWI climate model compared to the other two systems. This point is clarified in the text. In addition, since the grid used by AWI is not trivial to understand without a supporting plot, we are directing the reader to Fig. 4 of Sein et al. (2016)  **[pg. 4, l. 21–26]**.

12) Section 2.3 is written very clearly. It may be worth saying something about including SIV in the SEM. IT seems to me that SIV could dominate the results since the autocorrelation is so strong in SIV.
This is a good point. We have incorporated your suggestion to the last paragraph of Sec. 2.3  **[pg. 8, l. 24 to pg. 9, l. 2]**.

13) Section 2.4, numerous grammar issues in this section. This section is rough and needs revision.
To bring clarity to the text, we reviewed and rewrote the entire Section 2.4.

14) Fig. 10, the lag/lead time for the reconstruction is not clear to me, related to comments about Fig. 1.
As for Fig. 1, this is indeed a lag-0 comparison. This info is clarified in Sec. 3.2.2 [pg. 19, l. 7] and in the Fig. 10's caption.

15) Page 22, line 30, remove 'respectively'. I do not think that is needed here because the numbers are the same as highlighted by the word 'both'.
Indeed, "respectively" was removed from the text [pg. 25, l. 9].

---

## Author Response (AR2)

**Editor Decision: Publish subject to technical corrections (26 Jun 2020) by Petra Heil**

Comments to the Author: Dear authors. Thank you for the diligent technical revision, much appreciated.

Non-public comments to the Author:

However, your rewrite has impacted the readability of your ms severley. I have now accepted your ms - with some REALLY MINOR technical requests (see below). I strongly suggest that you ensure your text is straight forward for direct comprehension by the reader, even if you consult some English language editing service.

Dear Petra,

Thanks a lot not only for handling the review process of our manuscript but also for your very detailed recommendations and suggestions, which we have very much appreciated.

See below our answers to your comments. Please, note that we had made a few other minor editions in order to make the text straight forward for direct comprehension by the reader.

Yours sincerely and on behalf of all co-authors,

Leandro Ponsoni

 Pls unify your lat/lon details: Some show a ".0", some not: l1-12: (158.0 ° W, 79.5 ° N)
 VERSUS
 l1-13: (40 ° E, 88.5 ° N)
 --> Add ".0" after "40" and after "107" in line 1-13... and check throughout your ms. All latitudes and longitudes are now displayed with decimals.

l1-17: Change "if we look at" to "if we inspect". Replaced. Also in other instances throughout the manuscript.

l2-2: Change "continous" to "ongoing". Replaced.

l2-4: Correct "respective reduction" to "respective reductions" (plural). Corrected.

l2-5: Suggest to change "led to significant impacts at global and regional scales." to "had significant impact on climate processes at global and regional scales". Accepted suggestion.

l2-5: Remove "some" from "some aspects". Corrected.

l2-7: Change "the higher occurrence" to "the increased occurrence". Replaced.

l2-10: Change "are changing their foraging behavior" to "are adapting their foraging behaviour". Replaced.

l2-11: Remove "have" from "have experienced". Removed.

l2-16: Change "is exponentially increasing," to "is increasing exponentially," Corrected.

12-19: Change "the answer" to "there is clear anser".

The sentence "the answer to the question of whether or not this decline in sea ice is affecting the interannual variability of the pan-Arctic SIV, and the other way around, is not clear yet" was replaced by "there is no clear answer to the question of whether or not this decline in sea ice is affecting the interannual variability of the pan-Arctic SIV, and the other way around".

l2-20: Remove ", is not clear yet". Related to the change described above.

l2-20/21: This is no selfstanding sentence: "Although, recent model analyses suggest that this might be indeed the case (Van Achter et al., 2019)." Suggest to rewrite "Nevertheless, recent model analyses suggests the latter (Van Achter et al., 2019)." Corrected.

l2-21: Add "current" to read "current atmosphere-ocean general". Added.

l2-22: Change "more and more complex nowadays" to "are increasingly complex". Replaced.

l2-28: Change "To test the hypothesis" to "To test this hypothesis". Replaced.

l2-31: Change "we aim at inspecting" to "we investigate". Replaced.

l3-5: Change "these authors" to "they". Replaced.

l3-10: Change "records of observations" to "observational records". Replaced.

l3-11: Remove "the" to read "allowing statistical models". Corrected.

l3-11: Correct "on this data," to "on these data,". Corrected.

l3-13: Change "is not made available year-round" to "is mainly limited the winter months" Since sea ice thickness data from satellites are available during other months than winter (e.g., for some years ICESat provided data in October and November), this correction was implemented but in a slightly different way. The following sentence is added to the manuscript: "In situ measurements of sea ice thickness needed for calculating the SIV are far too expensive, while satellite campaigns such as ICESat (Kwok et al., 2007), CryoSat-2 (Kwok and Cunningham, 2015), and SMOS (Tian-Kunze et al., 2014; Kaleschke et al., 2016) present well-known limitations in the warmer seasons."

l3-14: What is a "classical satellite campaign(s)"? "classical" was dropped from text (see above).

l3-22: Change "the best in situ locations for sampling predictor variables" to "the best locations for in situ sampling of predictor variables". As suggested, the question (ii) was reformulated in the Introduction and Conclusion.

l4-6: Remove "In our study, we use two different model horizontal grids for each of the three models. Namely," and capitalize "the" --> "The". We incorporated this suggestion and moved the info that each model has two horizontal grid resolutions to the previous paragraph.

l4-22: Change "finer resolution" to "higher resolution". Changed.

l5-Fig1: Add "s" to "model snad remove "outputs" to read "the sixe models used in this work". We replaced "six model outputs" by "six model configurations". To avoid confusion, we didn't use "six models" since we are using only three different AOGCMs (two resolution each).

16-3 and 15 and onwards: "global" and "local" variables are not ideal terminology. Suggest to change "global variable" to "integrated variable" and remove "local" for the second type, to name it "variable" only.

Corrected.

l8-10: Pls explain, why the Error term falls away for SIV\_rec. The theoretical form of the linear regression has indeed an associated error. However, we do not know its true value in practice. To avoid confusion, we are crossing out the error term from Eq. 1.

l8-11: Change "To bring" to "To ensure". Changed.

19-4: Change "We intend too spot" to "We intend to identify". Replaced.

19-32: Change "we can look for" to "we search for". Replaced.

19-33: Change "is spotted" to "is identified". Replaced.

110-1: Change "near each other" to "in close proximity of each other". Replaced.

l11-10 onwards: Paragraphs are very short... only 2 or 3 sentences long. Can you pull some together?

Former paragraphs 1st, 2nd and 3rd were merged. The same the 4th and 5th paragraphs.

113-23: Add "ourselves" to read "reminding ourselves that only". Added.

113-25: Change "poor capacity" to "poor skill" or "poor capability". Replaced by "poor capability".

113-30: Change "The spotted ideal location" to "The resulting ideal location". Replaced.

l14-Fig6 caption: Change "the 1st optimal location" to "the primary optimal location". Changed.

114-6: This doesn't read well/make sense: "feed" in "feed with predictor variables". *"feed with predictor variables from"* was rewritten as *"using predictor variables from"*. Also in the abstract.

116-4: Change "Once a 1st common optimal site is determined, we fix it for all datasets and so look for the 2nd best location." to "Once the primary common optimal site has been identified and accepted for all datasets, we search for the 2nd best location.". Changed.

l16-4 to 6: Very short paragraph. Connect with next paragraph. Connected.

116-10: Improve style of "rom where it is separated by a distance of about 167 km". Please, see next answer.

116-11: Change "are placed at" to "are all identified within". We think this suggestion is changing the meaning of our analysis. Nevertheless, we are taking into account the criticism and rephrasing the entire sentence as follow:

"The 2nd site is about 167 km from the North Pole. The 3rd, 4th and 5th points are placed at the offshore domain of the Laptev Sea near the transition with the Central Arctic, in the Central Arctic to the north of the Canadian Islands, and within the Beaufort Sea, respectively."

116-14: Rewrite to improve English: "If we think of an optimal observing framework, in which only a few observational platforms are deployed,"

"If we think of an optimal observing framework, in which only a few observational platforms are deployed" was removed from the text.

118-10: Change "near-coast side of the Laptev Sea." to "near-coastal Laptev Sea." Replaced.

119-3: Rewrite to improve readbility: "Again, we will make use of the RMSE to evaluate". Rewritten as "We will make use of the RMSE to evaluate how good is our statistical prediction in terms of absolute values as in the previous sections, but..."

l20-5: Change "could be explained" to "may be explained". Replaced.

1120-6: Change "Also interestingly is he fact that for the R 2 metric, the opposite from" to "In addition, it is of interest that the R^2 metric behaves in oppoistion to". Changed.

123-5: Improve English in "For the sake of fairness," (replace). *"For the sake of fairness"* was dropped from the text.

l23-31 to 33: Very short paragraph. Merged with the previous paragraph.

125-1: Rephrase "That being said, we can recapitulate and". Rephrased as *"We now recall and objectively answer the first open question posed in the introduction of this manuscript:"*.

125-4: Change "If we take into account" to "Taking into account". Replaced.

l25-10: Change "is developed" to "had been developed". Replaced.

l25-11: Improve English of "Such observations could eventually be performed in the framework". Please, see next answer.

l25-12: Improve English of "So, we considered ...".

Rewritten as "For example, such a system could be part of an operational oceanography program in which predictor data are provided to the statistical model through in situ observations (e.g., oceanographic moorings and/or buoys) of SIT, SST, and Drift. The SIC and pan-Arctic SIA could also be incorporated into the statistical model since they are regularly sampled by satellites."

l25-15: Change "The OHT and the SIV are here disregarded as predictors." to "Here OHT and the SIV are disregarded as predictors." Replaced.

l25-15 to 20: Rewrite to improve style in "The former did not act as a skillful predictor, at least not when using monthly means. .... as follows:"

Rewritten as "The former is not a skillful predictor (as shown in Section 3.1), while the latter is the variable that we want to predict. We restricted our analyses to a maximum of ten optimal locations, although a reduced number of observational sites are sufficient to fairly reproduce the SIV anomaly. The results from Section 3.2 provide further elements to answer the other three open questions of this study, as follows:"

125-23: Rewrite for improved English: "We have here identified ten optimal locations." Rewritten as: *"The ten first best sampling locations were identified"*

l25-31: Change "what is also confirmed" to "that is also confirmed". Replaced.

126-3: Remove " by adding new sites" and replace with "the simulation". Rewritten as "Adding further to six well-placed locations does not substantially improve the statistical predictability". l26-8: Change "finer" to "higher". Replaced.

l26-11: Change "if we look at the" to "investigating the". Replaced. Also in other instances throughout the manuscript.

l26-12: Add "the" to read "by the numerical models". Corrected.

l26-17: Change "We envisage three main way by which this work could" to "This work has scope to positively impact three main approaches to".

Rewritten as "We believe that this work positively impacts three aspects of a real-world observing system.".

l26-32: Replace "be taken" with "be exercised". Replaced.

[revised manuscript text omitted]

35 km3 and  $2.56 \times 10^3$  km3 for AWI and HadGEM3, but much larger to ECMWF ( $26.17 \times 10^3$  km3). The standard deviations

(STD) from the SIV anomalies indicate that interannual variabilities are also higher for the coarser grid versions (Fig. 1b). The difference between coarser and finer higher resolutions for AWI, ECMWF, and HadGEM3 are  $0.30 \times 10^3$  km3,  $1.78 \times 10^3$  km3, and  $0.43 \times 10^3$  km3. We recall that the term anomaly in this work refers to the detrended and deseasonalized time series. In practical terms, the anomaly is calculated by excluding the individual trend provided by a second-order polynomial fit of each individual month.